# Direct and indirect impacts of the COVID-19 pandemic on life expectancy and person-years of life lost with and without disability: A systematic analysis for 18 European countries, 2020–2022

Sara Ahmadi-Abhari[1]*, Piotr Bandosz[2], Martin J. Shipley[3], Joni V. Lindbohm[3,4,5], Abbas Dehghan[1,6,7‡], Paul Elliott[1,6,7‡], Mika Kivimaki[4,8‡]

1 Department of Epidemiology and Biostatistics (EBS), School of Public Health, Imperial College London, London, United Kingdom, 2 Department of Preventive Medicine and Education, Medical University of Gdansk, Gdansk, Poland, 3 Department of Epidemiology and Public Health, University College London, London, United Kingdom, 4 Clinicum, Faculty of Medicine, University of Helsinki, Helsinki, Finland, 5 Broad Institute of MIT and Harvard, The Klarman Cell Observatory, Cambridge, Massachusetts, United States of America, 6 MRC Centre for Environment and Health, Imperial College London, London, United Kingdom, 7 United Kingdom Dementia Research Institute, Imperial College London, London, United Kingdom, 8 Brain Sciences, University College London, London, United Kingdom

‡ These authors are joint senior authors on this work.
* s.ahmadi-abhari@imperial.ac.uk

## Abstract

### Background

The direct and indirect impacts of the COVID-19 pandemic on life expectancy (LE) and years of life lost with and without disability remain unclear. Accounting for pre-pandemic trends in morbidity and mortality, we assessed these impacts in 18 European countries, for the years 2020–2022.

### Methods and Findings

We used multi-state Markov modeling based on several data sources to track transitions of the population aged 35 or older between eight health states from disease-free, combinations of cardiovascular disease, cognitive impairment, dementia, and disability, through to death. We quantified separately numbers and rates of deaths attributable to COVID-19 from those related to mortality from other causes during 2020–2022, and estimated the proportion of loss of life expectancy and years of life with and without disability that could have been avoided if the pandemic had not occurred. Estimates were disaggregated by COVID-19 versus non-COVID causes of deaths, calendar year, age, sex, disability status, and country. We generated the 95% uncertainty intervals (UIs) using Monte Carlo simulations with 500 iterations. Among the 289 million adult population in the 18 countries, person-years of life lost (PYLL) in millions were 4.7 (95% UI 3.4–6.0) in 2020, 7.1 (95% UI 6.6–7.9) in 2021, and 5.0 (95% UI 4.1–6.2) in 2022, totaling 16.8 (95% UI 12.0–21.8)

**Data availability statement:** This study used data from the English Longitudinal Study of Ageing (ELSA) and Survey for Health, Ageing and Retirement in Europe (SHARE). Data from ELSA are available from the UK Data Service (https://doi.org/10.5255/UKDA-Series-200011) under conditions of the End User License (https://www.elsa-project.ac.uk/accessing-elsa-data). Access to the data collected and generated in the SHARE projects is provided free of charge for scientific use globally, subject to European Union and national data protection laws as well as the Conditions of Use (https://share-eric.eu/data/data-access).

**Funding:** SAA was supported by institutional funding from Imperial College London and the European Institute of Innovation and Technology, EIT-Health (210920). PE was supported by the UK Medical Research Council (MR/S019669/1) and as a group leader in the Imperial College Dementia Research Institute was supported by the Medical Research Council, Alzheimer's Society and Alzheimer's Research UK. MK was supported by the Wellcome Trust, UK (221854/Z/20/Z), UK Medical Research Council (MR/R024227/1, MR/Y014154/1), and Academy of Finland (350426). JVL was supported by Academy of Finland (339568). The funders of the study had no role in study design, data collection, data analysis, data interpretation, or writing of the report. All authors had final responsibility for the decision to submit for publication.

**Competing interests:** The authors have declared that no competing interests exist.

**Abbreviations:** ELSA, English Longitudinal Study of Ageing; GBD, Global Burden of Disease; GDP, gross domestic product; LE, life expectancy; PYLL, person-years of life lost; SHARE, Survey for Health, Ageing and Retirement in Europe; STMFs, short-term mortality fluctuations; UIs, uncertainty intervals; YLL, years of life lost.

million. PYLL per capita varied considerably between the 18 countries ranging between 20 and 109 per 1,000 population. About 60% of the total PYLL occurred among persons aged over 80, and 30% in those aged 65–80. If the pandemic were avoided, over half (9.8 million (95% UI 4.7–15.1)) of the 16.8 million PYLL were estimated to have been lived without disability. Of the total PYLL, 11.6–13.2 million were due to registered COVID-19 deaths and 3.6–5.3 million due to non-COVID mortality. Despite a decrease in PYLL attributable to COVID-19 after 2021, PYLL associated with other causes of death continued to increase from 2020 to 2022 in most countries. Lower income countries had higher PYLL per capita as well as a greater proportion of disability-free PYLL during 2020–2022. Similar patterns were observed for life expectancy. In 2021, LE at age 35 (LE-35) declined by up to 2.8 (95% UI 2.3–3.3) years, with over two-thirds being disability-free. With the exception of Sweden, LE-35 in the studied countries did not recover to 2019 levels by 2022.

## Conclusions

The considerable loss of life without disability and the rise in premature mortality not directly linked to COVID-19 deaths during 2020–2022 suggest a potential broader, longer-term and partially indirect impact of the pandemic, possibly resulting from disruptions in healthcare delivery and services for non-COVID conditions and unintended consequences of COVID-19 containment measures. These findings highlight a need for better pandemic preparedness in Europe, ideally, as part of a more comprehensive global public health agenda.

## Author summary

### Why was this study done

- Estimating the person-years of life lost (PYLL) due to the pandemic broken down by cause of death and disability status is informative for policy setting and resource allocation in healthcare, but not thoroughly investigated

### What did the researchers do and find

- Many people who died during the pandemic would likely have lived longer if the pandemic had not happened. To quantify these lost years, we tracked incidence of disease, disability and mortality, and integrated data from multiple sources in a statistical model

- In 18 European countries constituting a population of 289 million, 16.8 million person-years of life were lost due to the pandemic in 2020–2022. About 60% of the PYLL would have been lived without disability if the pandemic had been avoided.

- Of the total PYLL, 3.6–5.3 million were due non-COVID causes of death and attributable to the pandemic's indirect impact on mortality. The PYLL in 2020 were mostly due to COVID-19 deaths and decreased between 2021 and 2022 parallel to vaccination roll out, but those due to non-COVID deaths continued to increase in most countries.

- PYLL varied considerably between the studied countries, ranging between 20 and 109 per 1,000 population. Countries with lower gross domestic product had higher PYLL per capita, a disproportionately higher disability-free PYLL, and a higher proportion of PYLL related to the pandemic's indirect impacts.

## What do these findings mean?

- The considerable proportion and increase over time in PYLL due to non-COVID deaths, point to the potential long-term impacts of the pandemic. The substantial proportion of PYLL without disability bring to light an instinctive underestimation of the pandemic's impact, especially on the older population.

- Findings suggest the pandemic worsened socioeconomic inequalities in premature mortality between countries and widened sex differences in life expectancy.

- Main limitations of the study were the lack of observed data on mortality rates by disability status during the pandemic necessitating reliance on data modeling strategies, and between-country differences in policies to ascertain COVID-19 as cause of death. Uncertainties in data modeling are reflected in the 95% uncertainty intervals accompanying all findings.

## Introduction

COVID-19 was declared a public health emergency of international concern from January 2020 to May 2023 [1]. With a death toll of just over 7 million by the end of 2024, the virus continues to mutate, change, and cause morbidity and mortality worldwide. A comprehensive evaluation and quantification of the mortality impact of the COVID-19 pandemic is crucial for planning and evaluating policy responses.

Several studies have investigated changes in life expectancy (LE) or years of life lost (YLL) in various countries and regions across the globe by comparing observed and expected mortality rates [2–11]. However, a nuanced evaluation disentangling person-years of life lost (PYLL) directly attributable to COVID-19 from those indirectly attributable to the pandemic due to excess mortality from other causes is mostly lacking. Delayed and reduced delivery of healthcare and services for conditions other than COVID-19 [12,13], and the pandemic containment measures such as lockdowns and social distancing, may have negatively affected population health [14,15]. These factors together with the unknown long-term health effects of COVID-19 may have contributed to increased premature mortality. PYLL attributable to these indirect impacts of the pandemic and how they have evolved over time has not been thoroughly explored.

Another less investigated mortality impact of the pandemic is an evaluation of PYLL with and without disability. Early pandemic reports highlighted that older individuals and those with underlying conditions and multi-morbidities were at a higher risk of COVID-19 morbidity and mortality [16–18]. This may have underestimated the pandemic's impact, which also involves disability-free PYLL across all age groups. To date, mortality rates by disability status are not available at national levels. However, estimates of YLL by disability can be modeled by combining mortality trends and relative risks of death by health and disability status.

We conducted a comprehensive evaluation of the direct and indirect effects of the pandemic, taking into account the uneven impact across countries, disaggregation by age and sex, as well as estimates of PYLL and changes in LE with and without disability to depict a more accurate picture of the mortality impact of the pandemic during 2020–2022 to inform decision-making, priority setting, and resource allocation in healthcare.

As data on changes in LE or PYLL beyond 2021 are scarce, we estimated the impact of the pandemic on changes in LE and PYLL from 2020 to 2022 using a previously developed and validated multi-state Markov model [19,20], which accounts for dynamic changes in LE, and enables decomposition of estimates of PYLL by age, sex, and disability status. This modeling strategy also enables estimation of both the direct and indirect pandemic impacts on changes

in LE and PYLL and investigation of their dependency on the countries' gross domestic product (GDP).

Previous studies that follow the standard burden of disease methodology and calculate YLLs by comparing age at death with standard LE at that age may overestimate the true magnitude of life-years lost as not all individuals expect ideal LE. In comparison, our modeling strategy accounts for the lower expected LE among those with disease or disability in calculations of PYLL. In addition, by incorporating changes in age-specific rates of mortality over time as well as changes in cardiovascular disease risk as a major driver of improved LE, we may be able to obtain more accurate estimates of PYLL than in previous reports. We focused on 18 European countries where availability of data allowed harmonization for comparability: Austria, Belgium, Czech Republic, Denmark, Estonia, France, Germany, Greece, Hungary, Italy, the Netherlands, Poland, Portugal, Slovenia, Spain, Sweden, Switzerland, and the United Kingdom.

## Methods

To calculate LE and PYLL with and without disability due to the COVID-19 pandemic over the course of 2020–2022, we used a probabilistic discrete-time Markov model to track the transitions of the population aged 35 and over between eight health-states of disease-free, combinations of cardiovascular disease, cognitive impairment, dementia, and disability, through to death (Fig 1). The Markov modeling strategy accounts for improvements in cardiovascular disease risk as a driver of improved mortality rates and LE, based on pre-pandemic trends. To estimate PYLL by disability status, the Markov models incorporate trends in cardiovascular disease and dementia as, notwithstanding variations between countries, about a fourth of the prevalence of disability occurs as co-morbidity with cardiovascular diseases and over half of the prevalence of disability occurs as co-morbidity with dementia. For computational feasibility, disability associated with other causes such as diabetes, cancers, respiratory conditions, and others, which together account for the remaining fourth of the prevalence of disability, are entered in the model as an aggregate health state (state 8, Fig 1).

Initially populated using age- and sex-specific population numbers and prevalence estimates, calendar-year specific transition probabilities were applied at each 1-year iteration of the Markov models to predict numbers of deaths and prevalence of each health state at subsequent calendar years, in addition to LE and PYLL with and without disability. Country-specific input data to inform the model included the population numbers, the age- and sex-specific initial prevalence of each health-state in the model, and age-, sex-, and calendar-time-specific transition probabilities between health/disability states and to death.

### Sources of data

**Population numbers and mortality rates.** We obtained population numbers stratified by age and sex in each country at the model's starting year, along with observed mortality rates between 1998 and 2022 from short-term mortality fluctuations (STMFs) calculations on mortality.org [21]. Observed COVID-19 deaths from 2020 to 2022 were obtained from official statistics reported to the World Health Organization [22]. Age- and sex-specific COVID-19 mortality rates were available for the UK from the UK Office for National Statistics, and for other countries from the demography of COVID-19 deaths database managed by the French Institute for Demographic Studies (INED [23]). Further details are provided in S1 Appendix Methods (pp 1–13).

**Prevalence and transition probabilities for health states.** To calculate age- and sex-specific prevalence estimates and transition probabilities in the model for the United

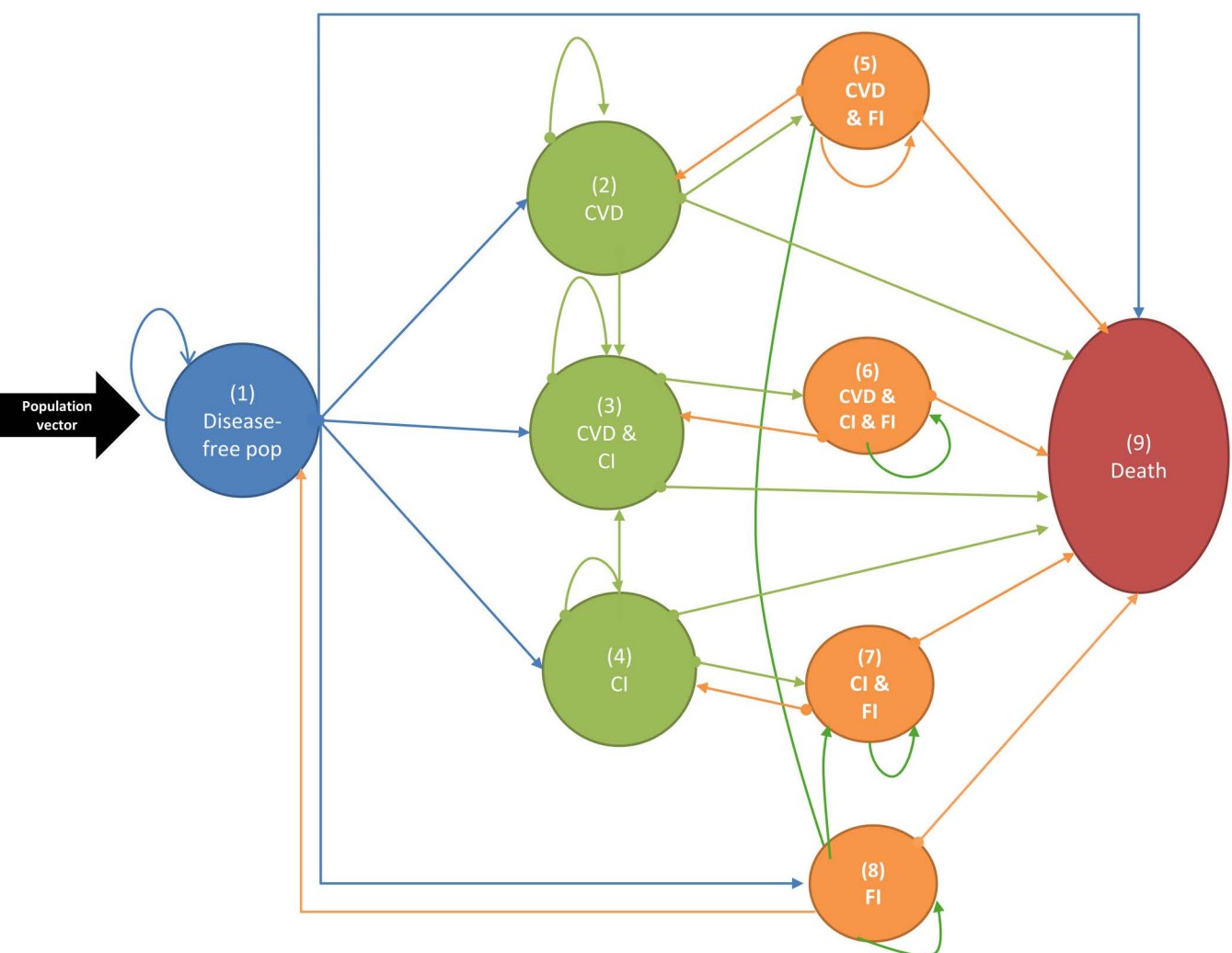

**Fig 1. Markov model structure.** CVD, cardiovascular disease; CI, cognitive impairment; FI, functional impairment (disability). Population vector represents the population reaching age 35 at each year and entering the model. The blue circle represents the population free of cardiovascular disease, cognitive and functional impairment or dementia. The orange circles represent the functional impairment (disability) status defined as inability to independently perform one or more basic activities of daily living. Dementia is defined as coexistence of cognitive and functional impairment (states 6 and 7). Disability associated with conditions other than CVD and dementia, such as diabetes, cancers, respiratory conditions, and other, are represented by state 8.

Kingdom, data from the English Longitudinal Study of Ageing (ELSA) waves 1–8 (2002–2017) were used as previously reported [24]. To enable comparability, for other countries, we used data from the first seven waves (2004–2017) of the Survey for Health, Ageing and Retirement in Europe (SHARE) [25], which follows standard and similar data collection structures as ELSA across all the studied countries. Further details of the study methods, data collection protocols, and case definitions in SHARE and ELSA are provided in S1 Appendix (pp 1–2). We included all countries with at least two waves of data collected, resulting in the listed 18 countries.

## Case definition for health states

Cardiovascular disease in SHARE and ELSA was ascertained by self-reported doctor diagnosis of myocardial infarction, stroke, angina, or coronary artery bypass grafting, or death from

cardiovascular causes [24,25]. Incident cardiovascular disease was defined as the first ever record of disease or attributable intervention.

Cognitive impairment was defined as impairment in 2 or more domains of cognitive function based on in accordance with the cognitive-impairment no-dementia [26] criteria based on standard cognitive tests conducted in the ELSA and SHARE studies [24,25]. Dementia was defined as co-occurrence of cognitive and functional impairment, or a self-reported doctor diagnosis of dementia as described in S1 Appendix Methods (p 2).

### Assessment of functional impairment (disability)

Participants of ELSA and SHARE or their proxy informants were requested to provide information about the ability of the participant to independently conduct basic activities of daily living (ADL) [24,25]. The ADLs are key tasks related to self-care and consist of getting in or out of bed, walking across a room, bathing or showering, using the toilet, dressing, cutting food and eating. Impairment in independently performing one or more activities of daily living lasting for more than 3 months was defined as functional impairment. We defined disability as functional impairment in this study, as it interferes with individuals' ability to live independently.

### Statistical analysis

**Markov models.** A detailed description of the statistical analyses in the development of the Markov models, including the mathematical formula, is provided in S1 Appendix Methods (pp 3–14). The definitions and assumptions underlying the Markov models are presented in Table A in S1 Appendix (pp 15–16). These methods and assumptions have previously been validated for the England and Wales model [19,20,27].

In each 1-year model iteration, calendar-year specific transition probabilities were imposed on age and sex specific prevalence of each health state in the model to estimate expected prevalence of disease and mortality in the next year. Details of the mathematical calculations are presented in S1 Appendix Methods (pp 3–8). Age-, sex-, and country-specific prevalence estimates at the baseline year of the Markov models were calculated using SHARE and ELSA data as described in S1 Appendix (p 9) and presented in Tables B and C in S1 Appendix (pp 17–18). Transition probabilities are displayed in Figs A and B in S1 Appendix (pp 19–20).

The PYLL due to the pandemic related excess deaths in 2020–2022 were defined as the years of life that would have been expected to be lived if the pandemic had not happened. We therefore ran and compared outputs from three Markov models. *Model 1:* To estimate person-years of life with and without disability that would have been expected to be lived in the absence of the pandemic, we included transition probabilities and expected mortality rates following pre-pandemic trends as model inputs. *Model 2:* To estimate loss in years of life due to registered COVID-19 deaths, we adjusted expected mortality rates from Model 1 for the additional registered deaths due to COVID-19 by age and sex for the years 2020–2022. *Model 3:* To estimate total loss in person-years of life due to deaths from any cause that would not have been expected under pre-pandemic trends, we applied mortality rates by disease and disability status that were derived from observed all-cause mortality rates by age and sex for the years 2020–2022. The difference in person-years of life between Models 2 and 3 with Model 1 provide an estimate for PYLL due to COVID and non-COVID deaths based on numbers of years that would have been expected to be lived with and without disability if premature deaths directly or indirectly related to the pandemic had not occurred and the pre-pandemic trends had continued. Further details are provided in S1 Appendix (pp 3–14).

To accurately estimate the expected prevalence of disability and mortality in the absence of the pandemic (Model 1), all transition probabilities entered in the Markov models were calendar-year specific following pre-pandemic trends in incidence of disease, disability, and mortality. The calendar effects for age- and sex-specific trends in incidence of cardiovascular disease, cognitive impairment/dementia, and functional impairment (disability), were quantified using ELSA and SHARE data. Further details are provided in S1 Appendix Methods (pp 12–13).

To obtain expected mortality rates in the absence of the pandemic to inform Model 1, all-cause mortality rates in each country were projected to the future by sex and 5-year age bands based on mortality rates observed between 1998 and 2019. Further details are provided in S1 Appendix Methods (pp 10–12), and Fig B in S1 Appendix (pp 20–25). For Model 2, to estimate loss of LE and PYLL explained by registered COVID-19 deaths, projected mortality rates from Model 1 were adjusted for the additional registered deaths due to COVID-19 for the years 2020–2022 (Table D in S1 Appendix (p 26)). Further details of data sources and calculation of COVID-19 mortality rates by age and sex for each country by year is provided in the S1 Appendix Methods (pp 10–11). In Model 3, observed all-cause mortality rates by country, age group, and sex for the years 2020–2022 obtained from STMF [21] were applied as described in S1 Appendix (pp 11–12). The mathematical formula for calculating PYLL from comparison of Models 2 and 3 with Model 1 are presented in S1 Appendix Methods (pp 3–8).

**Analyses by disability status.**  To estimate age- and sex-specific mortality rates by disability status that were required as inputs to the Markov models, we first obtained, from the SHARE and ELSA data, the hazard ratio of death from each disease and disability state compared to overall mortality rate in that age, and sex strata (Table E in S1 Appendix (pp 27–29)). We applied these hazard ratios to projections for expected all-cause mortality rates, and observed all-cause, COVID-related, and not COVID-related mortality rates to obtain estimates for those by disease and disability status. This method assumes the hazard of death by disability remains proportional to the baseline hazard of death over time by age and sex (further details are provided in S1 Appendix (pp 11–12)). In Fig C in S1 Appendix (p 30) we demonstrate mortality rates by disability status derived from this method. According to our calculations, among all studied countries, on average over half of excess deaths in 2020–2022 (i.e., difference in observed all-cause mortality from that expected under pre-pandemic trends) was in persons with disability (Table F in S1 Appendix (p 31)). By age-group, the proportion of excess deaths that we estimated to occur among persons with disability by applying the proportional hazards assumption was 65% in persons aged 80 + , 44% in those aged 65–79, 24% in those aged 50–64, and 11% in persons aged 35–49. The figures varied by country as demonstrated in Table F in S1 Appendix (p 31). To evaluate the reliability of this assumption, we compared model outputs with observed all-cause and COVID-19 mortality rates by disability status over the pandemic years in the UK [28] where such data were available. The comparison showed the proportional hazards method to derive mortality rates by disease and disability is unlikely to underestimate mortality rates by disability status (further details are presented in S1 Appendix Methods (pp 11–12), and Fig D in S1 Appendix (p 32).

LE, disabled LE, and disability-free LE were calculated from model outputs according to the Sullivan method [29]. PYLL with and without disability were calculated from the Markov models as described in S1 Appendix Methods (pp 3–13).

**Analysis by cause of death (COVID/non-COVID).**  We found, at varying time points in several countries, that the additional deaths attributed to COVID-19 were offset by lower than expected non-COVID mortality according to projected pre-pandemic trends. This may partly be due to reductions in non-COVID mortality rates resulting from pandemic control policies

that also result in reduced mortality from other causes with shared prevention strategies, such as flu and respiratory conditions; and partly due to replacement of inevitable non-COVID mortality by COVID-19 as the registered cause of death in the event of co-occurrence of morbidities. As sufficient data to validly distinguish between the two possibilities were not available, we conducted analyses based on two extreme scenarios to obtain the interval encompassing the proportion of PYLL by cause, assuming the true estimate lies between the two extreme scenarios. In *Scenario 1*, we assumed all the observed lower than expected non-COVID mortality rates are due to a true reduction in non-COVID deaths, and in *Scenario 2* we assumed all the observed lower than expected non-COVID mortality rates are due to replacement of expected non-COVID mortality by COVID-19 on cause of death registrations.

**Probabilistic sensitivity analysis.** To explore the impact of parameter uncertainty on model outputs, we conducted a probabilistic sensitivity analysis using Monte Carlo simulation. The procedure entails sampling from specified distributions for the input parameters that were used in the model for each data cycle. We calculated 500 iterations to estimate 95% uncertainty intervals (UIs) for output variables. Uncertainty in model parameters increases over time and varies between countries. Further details of the probabilistic sensitivity analysis can be found in S1 Appendix Methods (pp 13–14) and the statistical formula are provided in S1 Appendix Methods (pp 3–8).

**Association of person-years of life lost with GDP and vaccination coverage.** We examined the association of PYLL with the countries' GDP per capita and vaccination coverage.

*National GDP per capita*: This standard measure of a nation's economic output and added value created through the production of goods and services was obtained from the World Bank's world economic indicators report [30].

*Vaccination coverage:* Total doses of COVID-19 vaccine per 100 adult population were used as an indicator of vaccination coverage [31].

Stata-17 (StataCorp. College Station, TX: StataCorp LP) was used for data management and regression analysis to derive model inputs. A bespoke R-package was previously developed by author PB to implement the Markov models.

## Results

### Person-years of life lost (PYLL)

Among the 289 million population aged 35 and above in the 18 European countries, we estimated a total of 16.8 (95% UI 12.0, 21.8) million person-years of life were lost during 2020−2022, corresponding to 58.2 PYLL per 1,000 population. This figure consists of 4.7 (95% UI 3.4, 6.0) million PYLL in 2020, followed by 7.1 (95% UI 6.6, 7.9) and 5.0 (95% UI 4.1, 6.2) million PYLL in 2021 and 2022, respectively (Table 1). There was a total of 1.2 million COVID-19 deaths registered in these countries during 2020−2022 (441,000, 516,000, and 285,000 COVID-19 deaths in 2020, 2021, and 2022, respectively).

Total PYLL per 1,000 population was highest in Estonia (108.9), Poland (108.0), and Spain (100.3) and lowest in Switzerland (26.4), Denmark (20.7), and Sweden (19.6) as displayed in Fig F in S1 Appendix (pp 35–36).

Lower than expected mortality rates from non-COVID causes were observed in several countries, including Sweden, France, Italy, Belgium, Switzerland, Denmark, and the UK at various time points between 2020 and 2022 (Fig E in S1 Appendix (p 35)). This observation could have resulted from a true reduction in non-COVID mortality (Scenario 1), replacement of expected non-COVID mortality by COVID-19 on cause of death registrations (Scenario 2), or a combination of both. Out of the 16.8 million PYLL, based on Scenarios 1 and 2, 11.6–13.2

**Table 1. Person-years of life lost (PYLL) in the population aged 35 and over by year and cause of death, in thousands (000)[*].**

| | Measure of PYLL | Total 2020–2022 (000) | By year | | |
| --- | --- | --- | --- | --- | --- |
| | | | 2020 (000) | 2021 (000) | 2022 (000) |
| **All 18 countries** | | | | | |
| *Men and women* | | | | | |
| | Total | −16,819 (−21,778; 11,965) | −4,693 (−6,009; −3,378) | −7,115 (−7,854; −6,572) | −5,011 (−6,156; −4,095) |
| *Scenario 1: True reduction in non-COVID mortality[**]* | | | | | |
| Due to COVID deaths | | −13,192 (−18,305; −10,755) | −4,395 (−6,058; −2,731) | −6,009 (−6,687; −5,545) | −2,788 (−3,746; −2,064) |
| Due to other causes of death | | −3,626 (−6,624; 1,964) | −299 (−1,788; 1,191) | −1,105 (−1,609; −602) | −2,222 (−3,042; −1,402) |
| *Scenario 2: Expected non-COVID mortality replaced with COVID-19 as registered cause of death[**]* | | | | | |
| Due to COVID deaths | | −11,565 (−16,853; −8,436) | −3,811 (−5,134; −2,547) | −5,122 (−5,949; −4,927) | −2,632 (−3,607; −2,003) |
| Due to other causes of death | | −5,254 (−4,926; −3,529) | −882 (−2,148; 481) | −1,993 (−2,253; −1,298) | −2,379 (−3,116; −1,501) |
| **By Sex** | | | | | |
| *Men* | | | | | |
| | Total | −9,872 (−12,776; −6,901) | −2,813 (−3,997; −1,628) | −4,222 (−4,694; −3,924) | −2,837 (−3,550; −2,333) |
| *Scenario 1: True reduction in non-COVID mortality* | | | | | |
| Due to COVID deaths | | −6,636 (−9,529; −5,009) | −2,421 (−3,924; −919) | −2,924 (−3,319; −2,719) | −1,292 (−1,824; −970) |
| Due to other causes of death | | −3,235 (−5,205; 53) | −392 (−1,735; 952) | −1,298 (−1,549; −1,047) | −1,546 (−1,958; −1,133) |
| *Scenario 2: Expected non-COVID mortality replaced with COVID-19 as registered cause of death* | | | | | |
| Due to COVID deaths | | −6,297 (−9,178; −4,291) | −2,222 (−3,377; −851) | −2,803 (−3,183; −2,607) | −1,272 (−1,800; −954) |
| Due to other causes of death | | −3,575 (−3,598; −2,610) | −591 (−1,869; 687) | −1,419 (−1,666; −1,172) | −1,565 (−1,977; −1,154) |
| *Women* | | | | | |
| | Total | −6,946 (−9,390; −4,538) | −1,880 (−2,669; −1,092) | −2,893 (−3,213; −2,692) | −2,173 (−2,666; −1,820) |
| *Scenario 1: True reduction in non-COVID mortality* | | | | | |
| Due to COVID deaths | | −6,556 (−9,072; −5,339) | −1,973 (−2,982; −965) | −3,086 (−3,425; −2,874) | −1,497 (−1,982; −1,152) |
| Due to other causes of death | | −390 (−1,885; 2,377) | 93 (−805; 992) | 193 (−14; 399) | −677 (−1,025; −328) |
| *Scenario 2: Expected non-COVID mortality replaced with COVID-19 as registered cause of death* | | | | | |
| Due to COVID deaths | | −5,267 (−7,785; −3,530) | −1,589 (−2,366; −811) | −2,318 (−2,600; −2,156) | −1,360 (−1,800; −1,060) |
| Due to other causes of death | | −1,680 (−1,604; −1,008) | −291 (−1,074; 491) | −575 (−757; −393) | −813 (−1,140; −487) |
| **By Country** | | | | | |
| *Austria* | Total | −350 (−444; −244) | −88 (−92; −85) | −123 (−140; −116) | −138 (−165; −124) |
| *Belgium* | Total | −319 (−403; −241) | −147 (−166; −128) | −74 (−88; −61) | −98 (−121; −82) |
| *Czech Republic* | Total | −637 (−749; −540) | −180 (−186; −175) | −370 (−404; −346) | −87 (−136; −51) |
| *Denmark* | Total | −71 (−140; −19) | −4 (−4; −4) | −23 (−28; −22) | −43 (−52; −41) |
| *Estonia* | Total | −85 (−105; −67) | −12 (−12; −12) | −40 (−44; −38) | −32 (−39; −29) |
| *France* | Total | −1,108 (−1,647; −538) | −417 (−417; −417) | −325 (−398; −304) | −366 (−474; −338) |
| *Germany* | Total | −2,351 (−3,293; −1,320) | −338 (−527; −132) | −876 (−987; −865) | −1,136 (−1,333; −1,070) |
| *Greece* | Total | −508 (−648; −375) | −99 (−103; −94) | −245 (−270; −233) | −164 (−205; −143) |
| *Hungary* | Total | −593 (−764; −446) | −135 (−142; −128) | −355 (−389; −339) | −102 (−153; −75) |
| *Italy* | Total | −1,760 (−2,353; −1,219) | −731 (−841; −628) | −623 (−703; −549) | −407 (−543; −316) |

*(Continued)*

**Table 1.**  (Continued)

| | Measure of PYLL | Total 2020–2022 (000) | By year | | |
| | | | 2020 (000) | 2021 (000) | 2022 (000) |
|---|---|---|---|---|---|
| *The Netherlands* | Total | −371 (−581; −169) | −101 (−142; −57) | −160 (−184; −156) | −111 (−148; −102) |
| *Poland* | Total | −2,519 (−2,992; −2,054) | −763 (−853; −673) | −1,374 (−1,510; −1,283) | −382 (−579; −249) |
| *Portugal* | Total | −315 (−450; −149) | −108 (−138; −78) | −111 (−131; −106) | −96 (−125; −88) |
| *Slovenia* | Total | −99 (−128; −66) | −39 (−44; −33) | −38 (−44; −35) | −22 (−30; −18) |
| *Spain* | Total | −3,179 (−3,832; −2,456) | −583 (−723; −446) | −1,434 (−1,574; −1,363) | −1,162 (−1,397; −1,022) |
| *Sweden* | Total | −115 (−184; −47) | −56 (−71; −43) | −31 (−40; −28) | −28 (−40; −25) |
| *Switzerland* | Total | −140 (−188; −91) | −62 (−72; −51) | −35 (−43; −31) | −44 (−56; −39) |
| *United Kingdom* | Total | −2,299 (−2,879; −1,648) | −829 (−829; −829) | −877 (−1,001; −810) | −593 (−778; −495) |

*Numbers in parenthesis represent 95% uncertainty intervals obtained from Markov model Monte Carlo simulations with 500 iterations. Minus sign indicates life-years lost.

**Lower than expected mortality rates from non-COVID causes were observed in several countries at several time points. There are two scenarios that explain this finding: (1) Scenario 1: The observed lower than expected mortality rates from non-COVID causes is due to a true reduction in non-COVID deaths during the pandemic. At such time points, a portion of PYLL due to COVID-19 is compensated by the lower than expected non-COVID mortality. (2) Scenario 2: The observed lower than expected mortality rates from non-COVID causes are due to replacement of expected non-COVID mortality with COVID-19 on cause of death registrations in case of co-occurrence of morbidities. In this case, a substitution of causes for "expected" deaths has occurred and does not result in PYLL.

million PYLL (69% – 78%) were due to registered COVID-19 deaths. Conversely, 3.6–5.3 million PYLL (22% – 31%) were attributable to non-COVID mortality (Table 1). Country-specific estimates of PYLL by cause (COVID versus non-COVID) are shown in Figs 2 and 3, with details provided in Table G in S1 Appendix (p 37).

In nearly all studied countries, PYLL attributable to registered COVID-19 deaths decreased from 2021 to 2022 (Fig 2 and Table G in S1 Appendix (p 37)). The only exception was Denmark where no meaningful change was noted. In contrast, PYLL attributable to other causes of death continued to increase from 2020 to 2022 in most countries.

In all countries, total PYLL was higher in men compared to women (Figs 3 and F in S1 Appendix (pp 35–36). Combined, a total of 9.9 million (95% UI 6.9, 12.9) PYL were lost among men during 2020–2022 corresponding to 70.1 per 1,000 population. In contrast, 6.9 million (95% UI 4.5, 9.4) PYL were lost among women corresponding to 45.9 per 1,000 population (Tables 1 and 2). Men also experienced an about 1.5 times higher proportion of PYLL not explained by registered COVID-19 deaths compared to women (Table 1, Figs 3 and F in S1 Appendix (p 35–36)).

Older age groups experienced higher PYLL with 30% of total PYLL occurring in the population aged 65–80 and 60% in the 80 + age group (Fig 3 and Table 2). These age groups also experienced higher proportions of PYLL not explained by registered COVID deaths compared to those aged 35- < 65 (Fig 3 and Table 2).

If trends in cardiovascular disease, cognitive impairment, and functional impairment observed over the two decades prior to the pandemic had continued, avoiding the pandemic would have saved an estimated 9.8 (95% UI 4.7, 15.1) million PYL lived free of disability, accounting for 58% of the total PYLL during the pandemic (Fig 4 and Table H in S1 Appendix (p 39)). The pattern carried on to the older age groups (Fig 5 and Table 2). Nearly half the PYLL in the 80 + age group is estimated to have been lived without disability. The estimated number of disability-free PYLL was higher in the oldest compared to the younger age groups (Figs 5 and G in S1 Appendix and Table 2 (p 41)). The proportion of disability-free PYLL was higher among men (65%) than among women (49%). The estimated proportion of PYLL that was without disability was higher among lower-income countries, being highest in Greece (76%) and Hungary (73%) and lowest in Belgium (51%) and France (51%) (Fig 4 and Table H in S1 Appendix (p 39)).

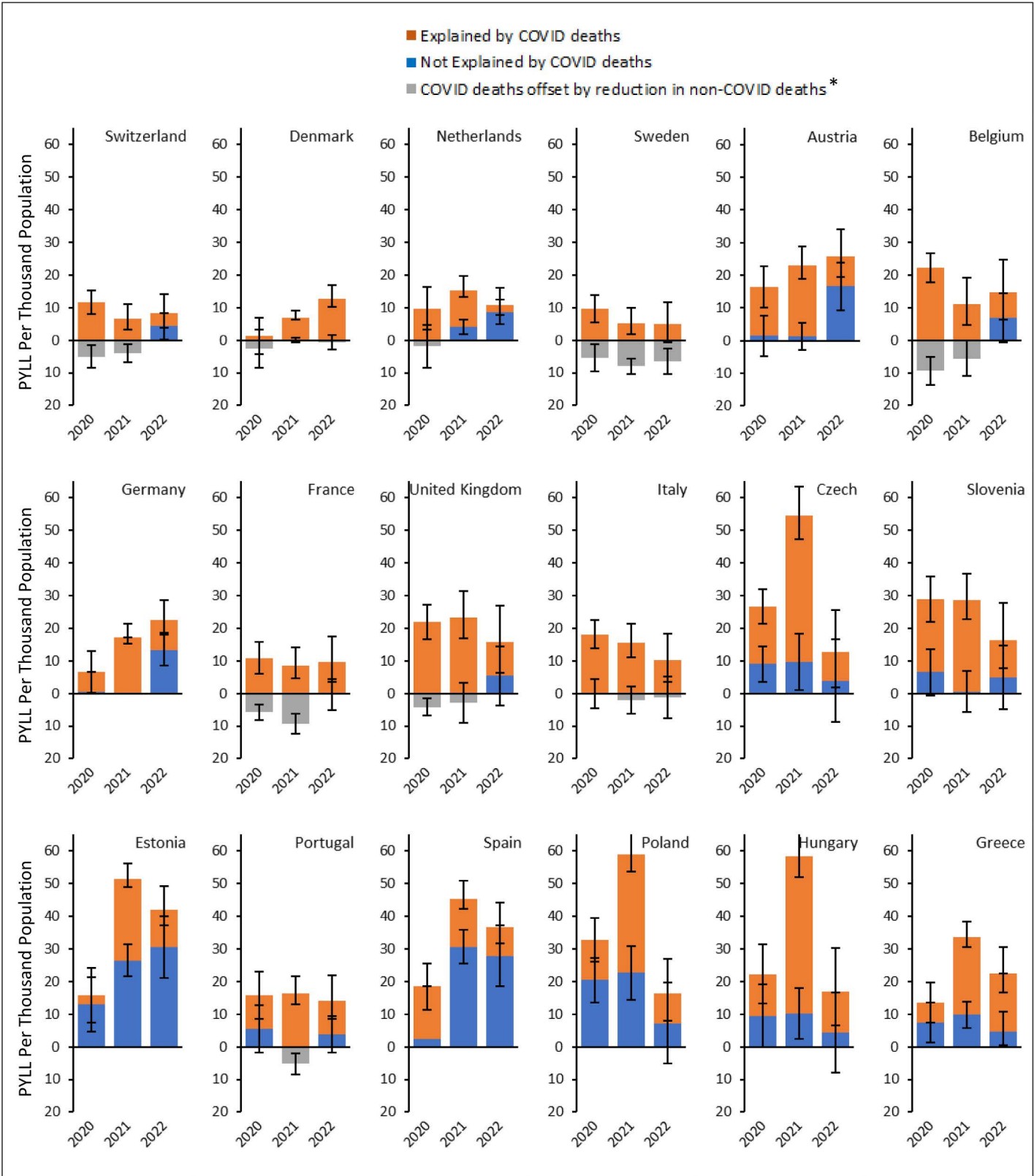

**Fig 2. Person-years of life lost (PYLL) per capita in the population aged 35 and over by year and country, sorted by descending gross domestic product per capita.** Gray bars represent the opposing effect of loss in person-years of life attributed to COVID-19 deaths that are compensated by the observed lower than expected deaths from non-COVID causes in several countries and time points, resulting in no net loss in person-years of life. There are two scenarios that explain this finding:

(1) If the observed lower than expected mortality rates from non-COVID causes is due to a true reduction in non-COVID mortality during the pandemic (Scenario 1), the PYLL due to COVID-19 would be the sum of the orange and gray bars. In this case, a portion of PYLL due to COVID-19 is compensated by the lower than expected non-COVID mortality (the portion displayed by the gray bars), reducing the total PYLL. (2) If the observed lower than expected mortality rates from non-COVID causes are due to replacement of expected non-COVID mortality by COVID-19 as registered cause of death in case of co-occurrence of morbidities (Scenario 2), the PYLL due to COVID-19 are represented by the orange bars. In this case, the gray bars are mere representations of substitution of causes for "expected" deaths and does not represent PYLL. (3) In both cases, PYLL due to non-COVID deaths are represented by the blue bars. Error bars represent 95% uncertainty intervals obtained from Monte Carlo simulation.

Total PYLL per capita was inversely and independently associated with the countries' GDP per capita and average doses of vaccine administered per adult population (Fig 6). Countries with higher GDP per capita and higher vaccination rates experienced lower PYLL due to both COVID and non-COVID causes (Fig H in S1 Appendix (p 43)). The deviation in the countries' PYLL from that explained by GDP per capita mirrored the average doses of administered COVID19 vaccines per adult population (Fig 6).

**Life expectancy at age 35 (LE-35).** Among the 18 studied countries, over the decade leading to the pandemic (2009–2019), LE at age 35 (LE-35) increased by an average of 1.2–4.2 months per year in men and 0.8–2.7 months per year in women (Fig I in S1 Appendix (p 44)). If pre-pandemic trends had continued, LE-35 was expected to increase slightly between 2019 and 2022. However, at the peak of the pandemic, LE-35 declined by up to 2.8 years (95% UI 2.3, 3,3; Figs 7 and 8 and Table I in S1 Appendix (p 46)) and over two-thirds of this decline was free of disability (Table I in S1 Appendix (p 46)).

Men had lower LE-35 in 2019 than women and the loss of LE during the pandemic was larger in men than in women, widening the sex gap in LE during the years 2020–2022 (Fig I in S1 Appendix (p 44), and Table I in S1 Appendix (p 46)). The cross-country gap in LE-35 also widened across 2020–2022, with countries having the lowest LE-35 and those with lower GDP per capita experiencing the greatest loss over this period (Fig J in S1 Appendix (p 53)). While the loss in LE-35 observed in 2020 partially recovered between 2021 and 2022, none of the 18 studied countries showed full LE-35 recovery by 2022 (Figs 7 and 8). Comparing observed and expected LE-35 for the year 2022, there was a deficit of 2 years or more in Estonia (men), and Spain (men); 1 year or more in Austria, Estonia (women), Germany, Greece, Portugal (women), and Spain (women), and a deficit of half a year or more in Belgium, Czech Republic, Denmark, France, Hungary, Italy, the Netherlands, Poland, Portugal (men), Slovenia, Switzerland, and the United Kingdom. Only Sweden recovered to the 2019 LE-35 levels having 0.3 years lower than expected LE-35 in 2022 (Table I in S1 Appendix (p 46)).

## Discussion

Our findings indicate that during the COVID-19 pandemic in 2020–2022, a total of 16.8 million years of life were lost in the population aged 35 years and older across 18 geographically and socioeconomically diverse European countries. Importantly, the majority of pandemic-related mortality losses in these countries affected individuals capable of leading independent lives without disability. In addition, there was a notable burden of PYLL not explained by registered COVID-19 deaths. Our data also shows that while no country escaped the pandemic's adverse impact, the loss in life years was more pronounced in countries with lower GDP per capita, such as Estonia and Poland, and smaller in those with higher GDP per capita, such as Switzerland, Denmark, and Sweden. This pattern also held true for LE, which dropped by up to 2.8 years in 2021, the worst year of the pandemic. The slowest recovery from this drop by 2022 was observed among men in Estonia and Spain, while the recovery was highest in Sweden.

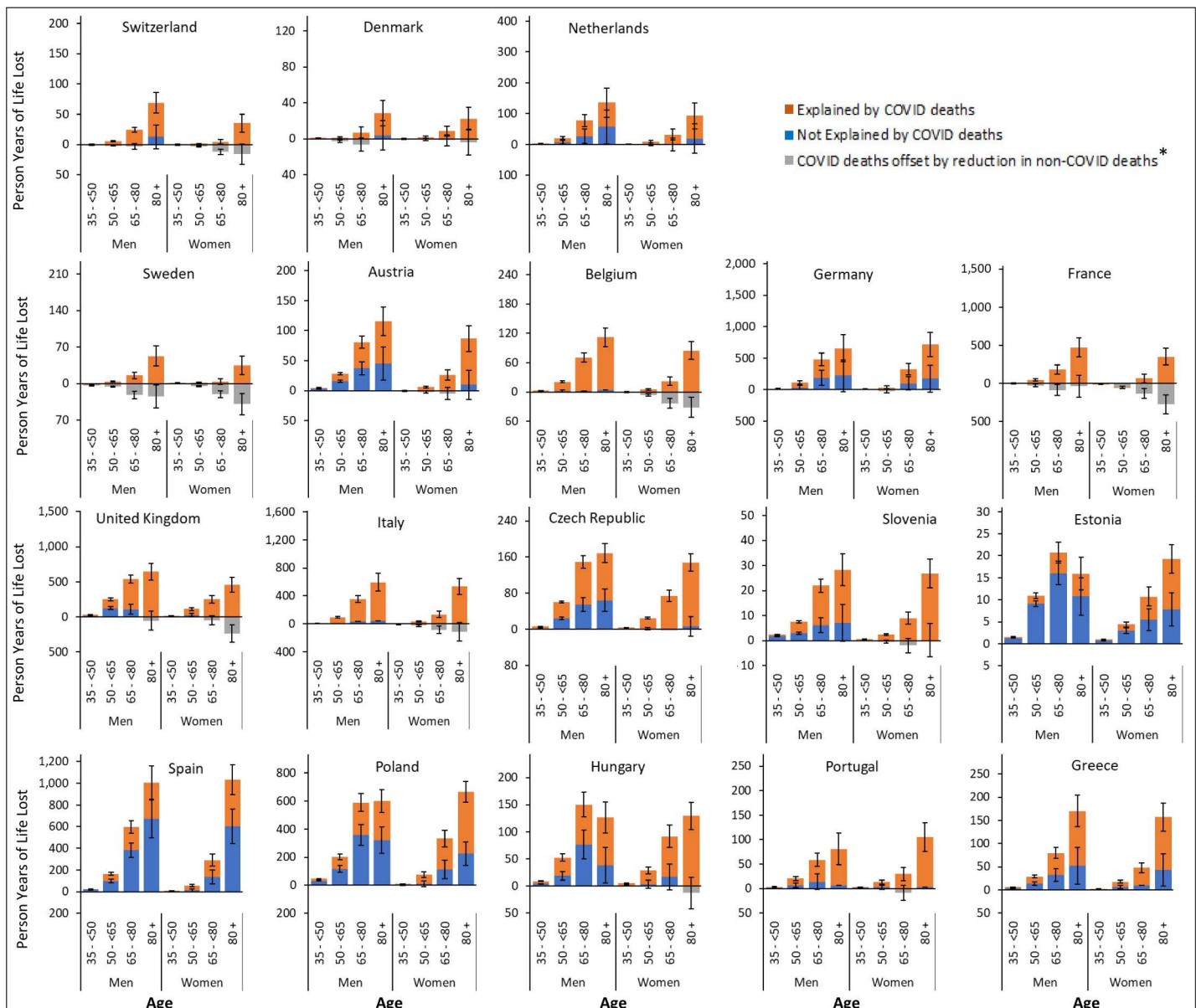

**Fig 3. Person-years of life lost (in thousands) in the population aged 35 and over by age-group, sex, and cause of death, sorted by descending gross domestic product per capita.** To enable comparability between countries, the range of the *Y*-axis is proportional to the total population aged 35+ in each country. Error bars represent 95% uncertainty intervals obtained from Monte Carlo simulation. Gray bars represent the opposing effect of loss in person-years of life attributed to COVID-19 deaths that are compensated by the observed lower than expected deaths from non-COVID causes in several countries and time points, resulting in no net loss in person-years of life. There are two scenarios that explain this finding: (1) If the observed lower than expected mortality rates from non-COVID causes is due to a true reduction in non-COVID mortality during the pandemic (Scenario 1), the PYLL due to COVID-19 would be the sum of the orange and gray bars. In this case, a portion of PYLL due to COVID-19 is compensated by the lower than expected non-COVID mortality (the portion displayed by the gray bars), reducing the total PYLL. (2) If the observed lower than expected mortality rates from non-COVID causes are due to replacement of non-COVID causes of deaths by COVID-19 due to co-occurrence of morbidities (Scenario 2), the PYLL due to COVID-19 are represented by the orange bars. In this case, the gray bars are mere representations of substitution of causes for "expected" deaths and does not represent PYLL. (3) In both cases, PYLL due to non-COVID deaths are represented by the blue bars.

Our estimates by disability status indicate that nearly 60% of the PYLL in the population aged 35 and above would have been lived without disability in the absence of the pandemic, a loss that may not have been fully recognized previously. While the

**Table 2. Person-years of life lost (PYLL) across years 2020–2022 by age group, cause of death, and disability, in thousands (000)\*.**

| | Age group | | | |
|---|---|---|---|---|
| | 35 to –<50 | 50 to –<65 | 65 to <80 | 80+ |
| *All 18 countries* | | | | |
| N in millions (000,000) | 97.9 | 96.7 | 67.5 | 27.1 |
| Total PYLL | −229 (−251; −207) | −1,557 (−1,735; −1,379) | −5,255 (−5,927; −4,582) | −9,778 (−11,191; −8,364) |
| PYLL *without* disability | −207 (−515; 121) | −1,316 (−2,548; −54) | −3,657 (−6,469; −1,012) | −4,612 (−7,292; −1,954) |
| PYLL *with* disability | −23 (−338; 271) | −256 (−1,352; 826) | −1,588 (−3,681; 532) | −5,131 (−7,340; −3,098) |
| *Scenario 1: True reduction in non-COVID mortality*\*\* | | | | |
| PYLL due to COVID deaths | −154 (−183; −149) | −1,163 (−1,397; −1,124) | −3,956 (−4,839; −3,804) | −7,918 (−9,828; −7,653) |
| PYLL due to other causes of death | −75 (−83; −44) | −394 (−454; −139) | −1,299 (−1,528; −338) | −1,860 (−2,288; 213) |
| *Scenario 2: Expected non-COVID mortality replaced with COVID-19 as registered cause of death*\*\* | | | | |
| PYLL due to COVID deaths | −127 (−154; −118) | −993 (−1,221; −912) | −3,453 (−4,305; −3,180) | −6,992 (−8,685; −6,233) |
| PYLL due to other causes of death | −102 (−97; −89) | −564 (−514; −467) | −1,802 (−1,622; −1,402) | −2,786 (−2,506; −2,132) |
| *By Sex* | | | | |
| *Men* | | | | |
| Total PYLL | −168 (−188; −148) | −1,134 (−1,297; −970) | −3,494 (−4,114; −2,874) | −5,076 (−6,336; −3,817) |
| PYL lost *without* disability | −153 (−343; 37) | −961 (−1,831; −104) | −2,542 (−4,232; −828) | −2,728 (−4,219; −1,243) |
| PYL lost *with* disability | −17 (−196; 166) | −185 (−901; 549) | −959 (−2,141; 225) | −2,339 (−3,354; −1,333) |
| *Scenario 1: True reduction in non-COVID mortality*\*\* | | | | |
| PYLL due to COVID deaths | −88 (−111; −80) | −688 (−872; −621) | −2,248 (−2,940; −1,986) | −3,611 (−4,961; −3,022) |
| PYLL due to other causes of death | −80 (−90; −55) | −446 (−532; −243) | −1,246 (−1,580; −482) | −1,465 (−2,200; 29) |
| *Scenario 2: Expected non-COVID mortality replaced with COVID-19 as registered cause of death*\*\* | | | | |
| PYLL due to COVID deaths | −79 (−99; −66) | −643 (−813; −544) | −2,112 (−2,773; −1,743) | −3,463 (−4,772; −2,603) |
| PYLL due to other causes of death | −89 (−89; −82) | −491 (−485; −426) | −1,382 (−1,341; −1,131) | −1,613 (−1,564; −1,214) |
| *Women* | | | | |
| Total PYLL | −61 (−77; −46) | −423 (−551; −296) | −1,761 (−2,235; −1,286) | −4,701 (−5,484; −3,918) |
| PYL lost *without* disability | −55 (−296; 194) | −356 (−1,240; 538) | −1,116 (−3,111; 868) | −1,884 (−3,853; 162) |
| PYL lost *with* disability | −6 (−251; 230) | −71 (−903; 735) | −629 (−2,300; 1,021) | −2,791 (−4,159; −1,453) |
| *Scenario 1: True reduction in non-COVID mortality*\*\* | | | | |
| PYLL due to COVID deaths | −66 (−82; −58) | −475 (−612; −416) | −1,706 (−2,223; −1,494) | −4,302 (−5,352; −4,147) |
| PYLL due to other causes of death | 5 (−5; 23) | 52 (−22; 203) | −55 (−322; 517) | −399 (−645; 741) |
| *Scenario 2: Expected non-COVID mortality replaced with COVID-19 as registered cause of death*\*\* | | | | |
| PYLL due to COVID deaths | −48 (−64; −37) | −354 (−473; −251) | −1,353 (−1,833; −1,001) | −3,554(−4,430; −3,052) |
| PYLL due to other causes of death | −13 (−12; −9) | −69 (−78; −45) | −408 (−402; −285) | −1,147 (−1,054; −866) |

\*Numbers in parenthesis represent 95% uncertainty intervals obtained from Markov model Monte Carlo simulations. Minus sign indicates life-years lost.

\*\*Lower than expected mortality rates from non-COVID causes were observed in several countries at several time points. There are two scenarios that explain this finding: (1) Scenario 1: The observed lower than expected mortality rates from non-COVID causes is due to a true reduction in non-COVID deaths during the pandemic. At such time points, a portion of PYLL due to COVID-19 is compensated by the lower than expected non-COVID mortality. (2) Scenario 2: The observed lower than expected mortality rates from non-COVID causes are due to replacement of expected non-COVID mortality with COVID-19 on cause of death registrations in case of co-occurrence of morbidities. In this case, a substitution of causes for "expected" deaths has occurred and does not result in PYLL.

overarching narrative has primarily focused on the higher risk of COVID-19-related deaths among people with underlying conditions and multimorbidities [16–18], our estimates suggest that over half of the PYLL in the 65 + age group and 47% of these years in the 80 + population would have been disability-free. In total, those aged 65 and above accounted for about 90% of the total PYLL, including both deaths from COVID-19 as well as those due to the pandemic's indirect impact. Our findings underscore the potential benefits of vaccinating the oldest individuals against COVID-19, and aligns with a

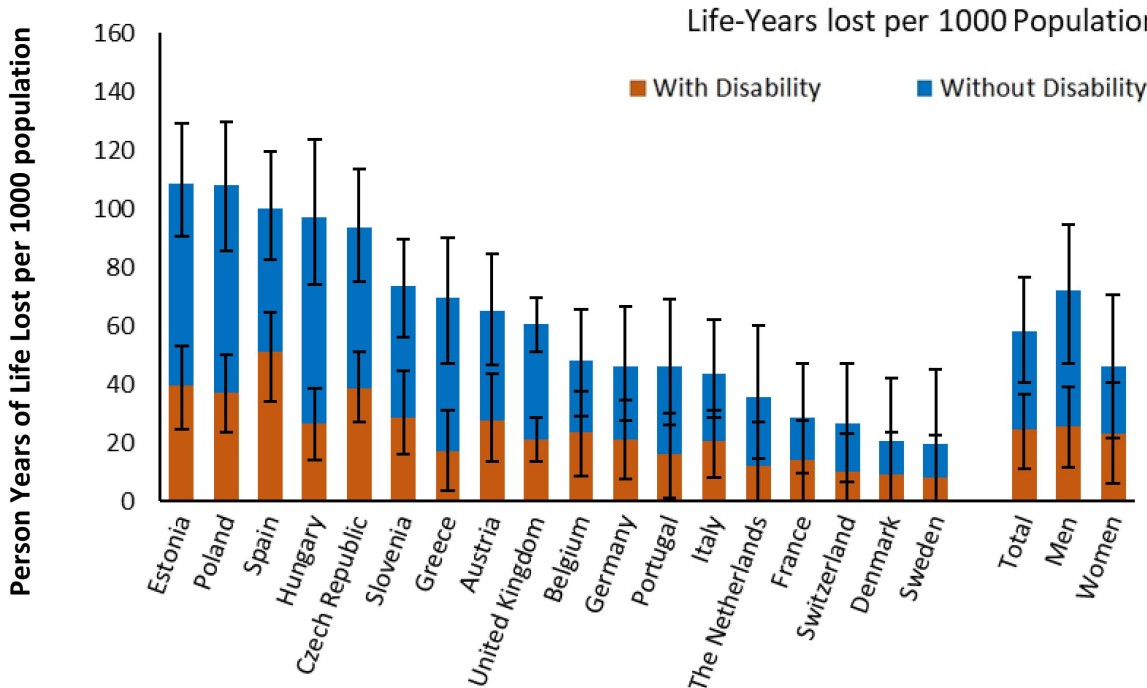

**Fig 4. Disabled and disability-free person-years of life lost per 1,000 population between ages 35 and 100 over the years 2020–2022.** Error bars represent 95% uncertainty intervals obtained from Monte Carlo simulation.

previous study suggesting vaccinating the oldest against COVID-19 could substantially save years of life [32].

A limitation of this study is the lack of observed mortality data by disability status during the pandemic. We estimated mortality by disease and disability over the pandemic years based on hazard ratios obtained in the preceding years. A concern with this method is that the pandemic may have disproportionately affected the rates of death among those with disability, especially among the older age groups. Assuming the hazard of mortality with disability to the hazard of all-cause mortality remains proportional over time, the estimated increases in mortality rates with disability would be a multiplication of the hazard ratio larger than increases observed in all-cause mortality by age and sex. We consequently estimated considerably larger increases in mortality rates with disability than without disability, especially in the older age-groups and in countries where excess mortality was higher in 2020–2022. Among the 18 countries, using this method, we estimated that on average over half (55%) of excess deaths were in persons with disability. By age group, we estimated 65% of excess deaths in the 80 + age group, 44% in the 65–79, 24% in the 50–64, and 11% in the 35–49 age groups were in persons with disability. The estimates of excess mortality by age and disability in Table F in S1 Appendix shows the proportional hazards assumption has likely captured the disproportionate effect of excess mortality amongst older people and those with disease and disability. Comparison of our mortality estimates for 2020–2022 for the UK with those obtained from the UK Office for National Statistics [28] presented in Fig D in S1 Appendix, indicates our estimates of mortality by disability based on the proportional hazard assumption are unlikely to have been underestimated.

We defined the PYLL due to excess mortality during 2020–2022 as the years of life that would have been expected to be lived if the pandemic had not happened. A large proportion of PYLL with disability is therefore accounted for by people who were living without disability

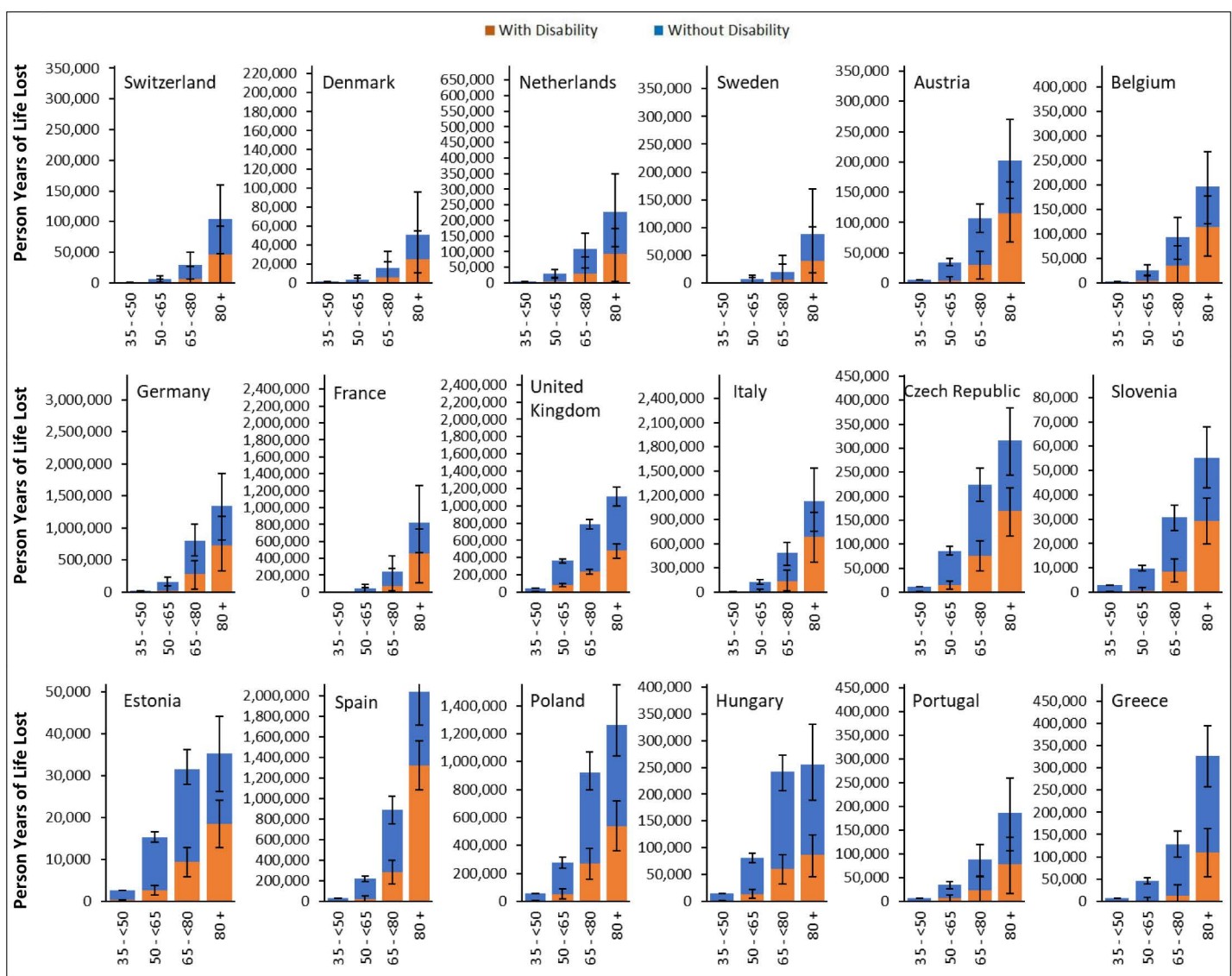

**Fig 5. Person-years of life lost with and without disability between ages 35 and 100 per 1,000 population by age-group, sorted by descending gross domestic product per capita.** To enable comparability between countries, the range of the *Y*-axis is proportional to the total population aged 35+ in each country. Error bars represent 95% uncertainty intervals obtained from Monte Carlo simulation.

at the time of death, but were predicted to have developed disability in the future if the pandemic had not resulted in premature death. As total mortality rates are captured by observed data, given over half of the excess mortality occurred in persons with disability in our calculations, and a prevalence of ~ 10% for disability at population level, it is unlikely that errors in estimation of mortality rates among those with disability would be large enough to materially change the conclusions of this study with regards to proportions of PYLL with and without disability.

There was a notable burden of PYLL not explained by registered COVID-19 deaths. Distinguishing between deaths attributable to COVID-19 and those unrelated is challenging due to variations in cause-of-death determination and registration systems across European countries [33]. Varying testing rates and differences in criteria for designation of COVID-19 as cause of

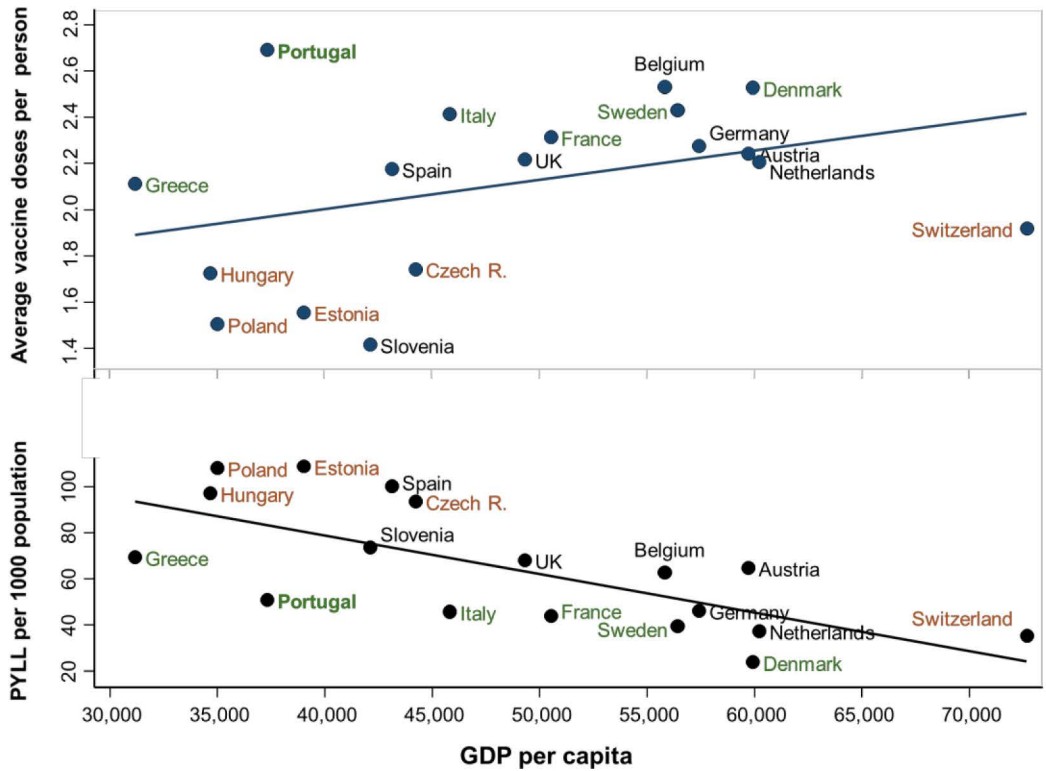

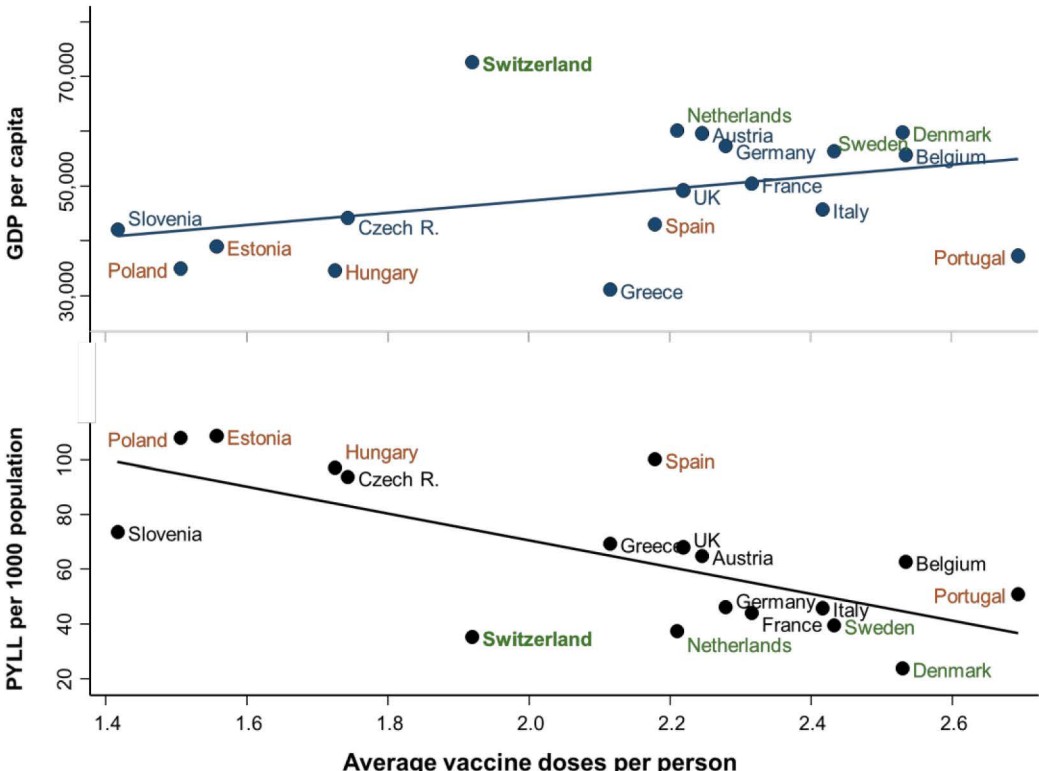

**Fig 6. Associations between person-years of life lost per 1,000 population over the years 2020–2022 with Gross Domestic Product per capita and vaccination coverage.**

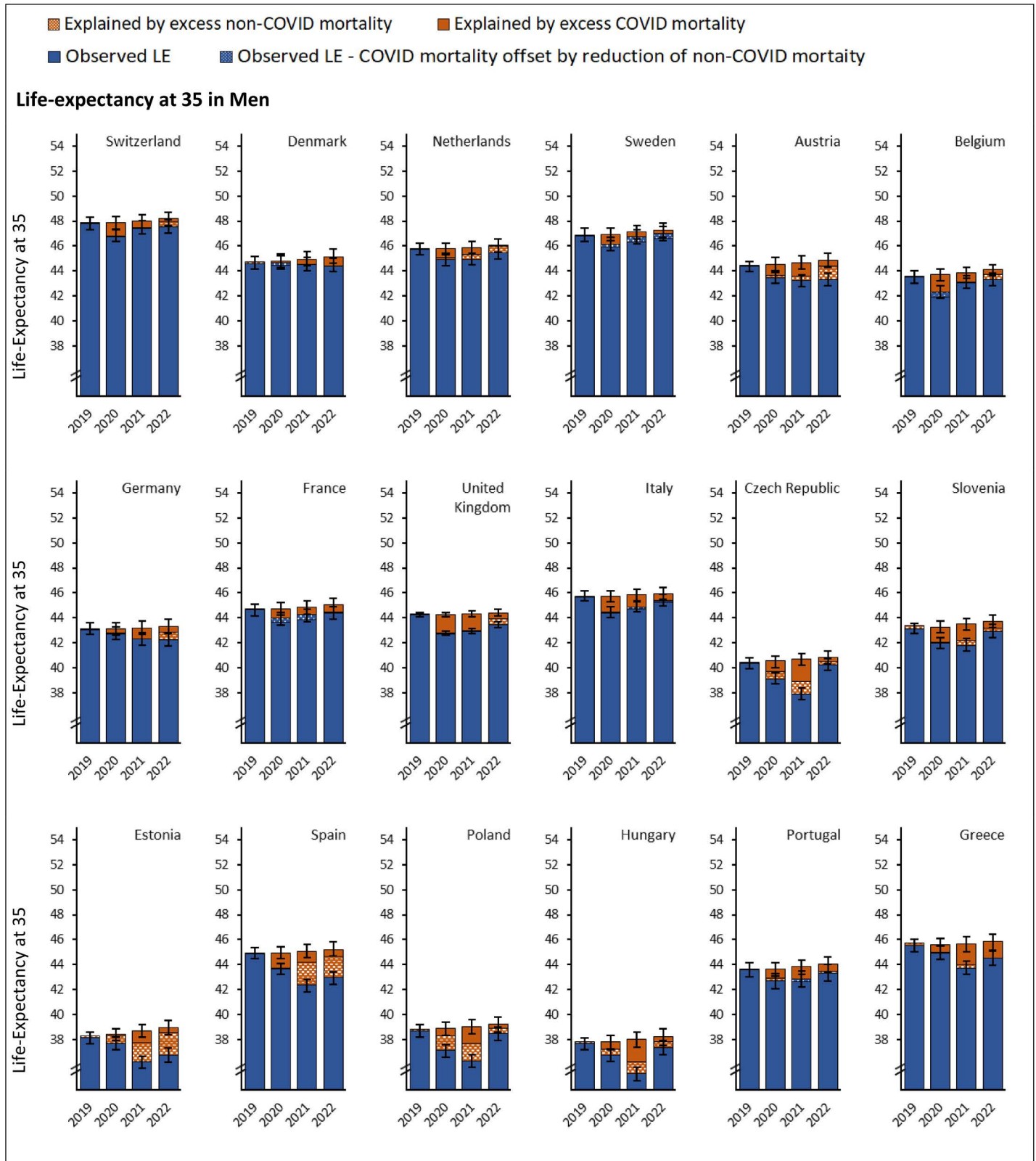

**Fig 7. Life-expectancy at age 35 in men by country, sorted by descending gross domestic product per capita.** Blue bars represent observed life expectancy at age 35. Orange bars represent difference between the observed and life expectancy expected under continuation of pre-pandemic trends.

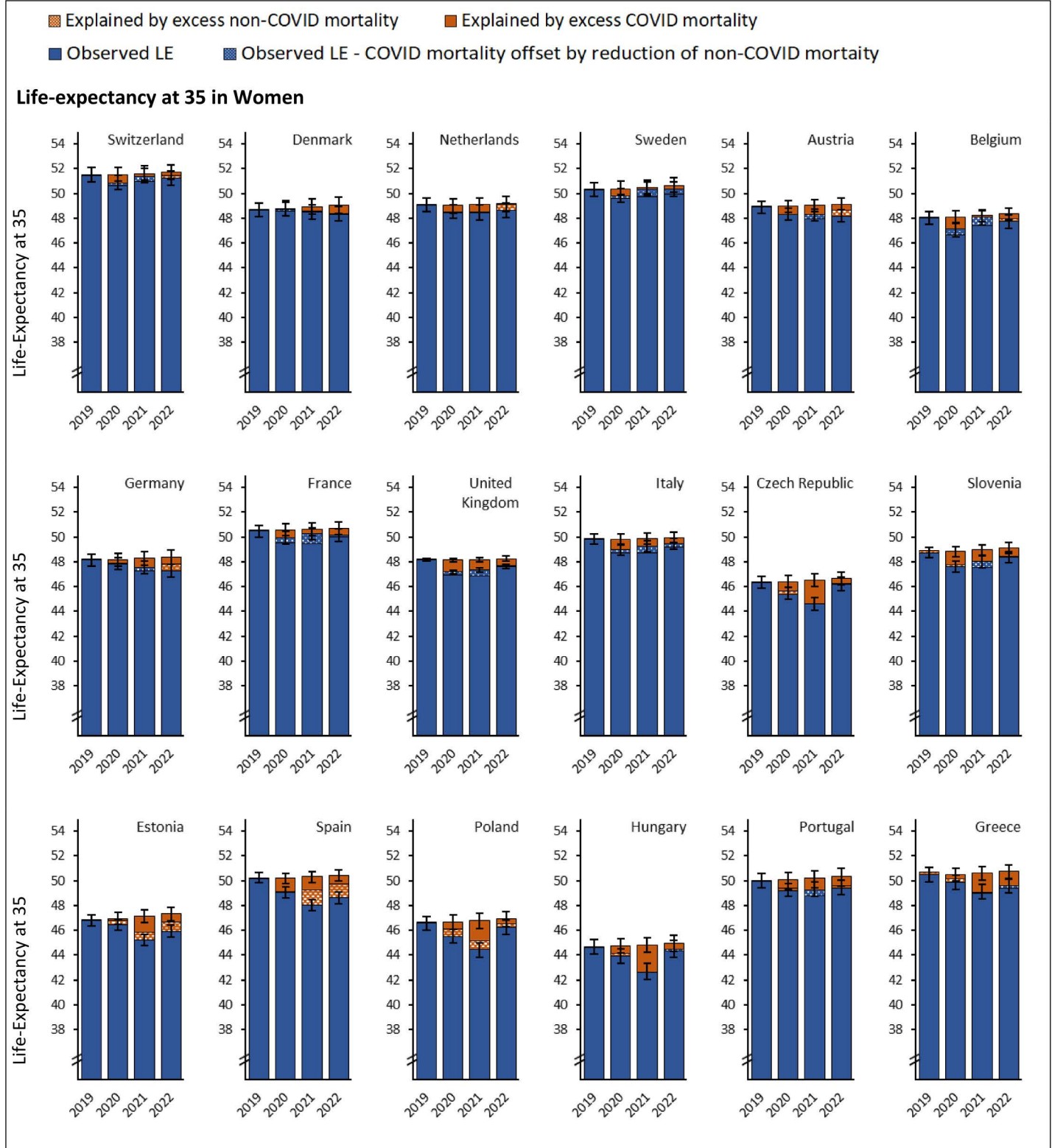

**Fig 8. Life-expectancy at age 35 in women by country, sorted by descending gross domestic product per capita.** Blue bars represent observed life-expectancy at age 35. Orange bars represent difference between the observed and life-expectancy expected under continuation of pre-pandemic trends.

death over time and location have complicated estimation of the direct and indirect impacts of the pandemic and cross-country comparisons. We observed lower than expected non-COVID mortality rates at different time points in Sweden, France, Italy, Belgium, Switzerland, Denmark, and the UK. This finding aligns with previous findings [2,34–36]. Several reports indicate the possibility that COVID-19 containment measures, such as wearing masks and social distancing, may be linked to reduction in other causes of death, such as the flu, pneumonia or infectious diseases which share similar prevention strategies [34–37]. On the other hand, non-COVID-19 deaths in patients with underlying terminal conditions, such as malignant cancers, may have been misclassified as COVID-19 deaths among those with a concurrent positive test [34–36]. Analysis of COVID-positive death certificates collected by the Italian National Statistics Institute in the municipality of Rome found other causes of death, such as cardiovascular diseases, malignant tumors, and diseases of the respiratory system, in 20% of the certificates [35]. Our observation of lower than expected non-COVID mortality rates could therefore be partly explained by overestimation of COVID-19 deaths or misclassification of deaths due to other causes as COVID-19 deaths, consequently underestimating non-COVID mortality. Moreover, countries employing broad definitions for COVID-19, including clinical symptoms and high-risk contacts without PCR test confirmation, may have led to an overestimation of COVID-19 deaths at certain time-points [36]. To address this challenge, we quantified PYLL using two extreme scenarios: (1) all reduction in non-COVID deaths was genuine and (2) all this reduction resulted from attributing some non-COVID deaths to COVID-19. Assuming the true impact lies between these two extreme scenarios, we estimate that on average between 22% (scenario 1) and 31% (scenario 2) of the excess life years lost during 2020–2022 were due to non-COVID mortality, with significant variation among countries.

PYLL in 2020 were mostly due to COVID-19 deaths and declined after 2021, coinciding with the introduction of vaccines. However, PYLL due to other causes continued to increase from 2020 to 2022 in most of the 18 countries. In 2022, nearly half of the PYLL were due to non-COVID deaths. As the pandemic progressed, most countries improved testing coverage, and expanded the definition and data sources for COVID-19 case of death determination, making under-registration of COVID-19 as cause of death unlikely to explain the increasing PYLL due to non-COVID mortality after 2021. These findings along with the higher proportion of non-COVID-related PYLL in older age groups, underscore the potential long-term implications of the pandemic's indirect impacts, including reduced or delayed provision of care and services for non-COVID-related conditions, as well as the effects of lockdowns and epidemic containment measures on lifestyle behaviors. These effects may have disproportionately affected older people and countries with lower GDP per capita [38].

We found that PYLL per capita were higher in countries with lower GDP per capita, and that these premature deaths were more likely to affect the population free of disability in these countries. Lower-income countries lost disproportionately higher PYLL not directly related to COVID-19. In contrast, higher-income countries had lower PYLL per capita, and a higher proportion of losses were with disability. Our findings indicate the indirect effects of the pandemic, as well as direct effects, may have negatively impacted lower-income countries in particular. Relatedly and consistent with previous findings [38–40], we found changing patterns of death over the pandemic have increased the gap in LE between countries, a factor that may warrant attention by international health organizations. Rates of PYLL in men were on average 1.5 times higher than those in women in all countries, a result consistent with the study by Pifarré and colleagues [5], which suggested a widening in the sex-gap in LE over the course of the pandemic.

We found COVID vaccination coverage was inversely associated with PYLL, independent of GDP. Several countries, notably Portugal, Italy, Denmark and Sweden, had lower PYLL

than countries with similar GDP per capita and the lower PYLL mirrored higher vaccination coverage in these countries. On the other hand, several countries including Poland, Estonia, Hungary, Czech Republic and Switzerland, had higher PYLL than countries with similar GDP per capita, also mirroring the lower vaccination coverage in these countries. COVID vaccination coverage was inversely associated with PYLL related to non-COVID deaths as well as those associated with COVID deaths, indicating that vaccination coverage may be a marker of the quality and reach of healthcare in the country, as well as the compliance of the population with government health policy and guidelines.

A limitation of this study is that the utilized data did not allow for the calculation of changes in LE at birth or PYLL before age 35. Given the near-exponential association of PYLL with age, with the 35 to <50 age group accounting for about 1% of PYLL, it is unlikely that including the younger age groups would significantly alter our conclusions. Another source of error is the accuracy of projections for future mortality rates which is inherently unpredictable. To partly account for uncertainty in projections in mortality and incidence of disease we accompanied our estimates with 95% UIs derived from Monte Carlo simulations.

To enable between-country comparisons, we included countries where the required input data for the Markov modeling strategy were available and collected following standardized and harmonized protocols. ELSA and SHARE studies have recruited broadly representative samples of the population aged 50 and over, and their cohabiting partners, and the data collection protocols were the same between countries. We only included countries if ELSA or SHARE data were collected at two or more time-points, resulting in the list of the 18 countries. There are between country differences in identification and registration of COVID-19 as cause of death between countries. However, it has previously been shown that the countries included in this study have reliable all-cause mortality registrations [33], ensuring the comparability of our estimates of total PYLL between countries. Although data sources followed similar data collection strategies, the UIs vary between countries due to differences in sample size of the studies and fluctuations in disease and mortality rates over time (further details are provided in the S1 Appendix Methods (p 14).

We have previously tested the validity of the Markov method in prediction of the prevalence of cardiovascular disease, dementia, and mortality in the UK against data obtained from independent studies [19,20,27] and in this analysis, compared mortality by disability status with data from the ONS for the UK. We were not able to conduct similar validation exercises for disease prevalence for other countries due to lack of data availability. It should be noted however that the data sources used in modeling data for all countries were harmonious, and although rates vary between countries, the shapes of the distributions and trends over time of the parameters were similar between countries. We therefore assumed that the Markov method would reasonably be applicable for other countries using country specific data that were collected from harmonized and standardized studies.

Life-years lost for excess deaths in 2020–2022 were calculated based on the counter-factual scenario in which the pandemic did not happen. Our modeling strategy accounts for changes in mortality, incidence of cardiovascular disease, cognitive impairment, and disability over time by incorporating calendar trends based on pre-pandemic trends in the calculations. We found that in all countries, the trends in mortality and incidence of disease and disability followed linear or log-linear trends over time over the two decades prior to the pandemic. We thus assumed the most likely scenario in the absence of the pandemic would have been the continuation of these trends. To obtain incidence and pre-pandemic trends in incidence of disease and disability, we used data from the ELSA and SHARE studies for the years 2002–2017 that were available at time of analysis. Although, it would have been ideal to use data that spans to the year 2019, incidence of disease and mortality have shown steady trends over time.

Therefore, obtaining pre-pandemic trends based on data collected to the year 2017 rather than 2019 is unlikely to have introduced a major bias in our findings. We accounted for uncertainty in future projections for input parameters, including the incidence of disease, disability and mortality rates, in Monte Carlo simulations.

We incorporated cardiovascular disease in the Markov models as it is a major driver for changing LE over time. Without accounting for declining mortality and cardiovascular disease incidence, the magnitude of PYLL may be underestimated. We included cognitive impairment and dementia in the models, as over 75% of cases of disability were associated with cognitive impairment, dementia, and cardiovascular disease. To balance model complexity with computational efficiency, we did not explicitly model other changes in morbidity, such as the increasing trend in type 2 diabetes, improved prognosis of several cancers, or respiratory conditions. We incorporated such changes by estimating a calendar trend for disability associated with causes other than cardiovascular disease and dementia as an aggregate state. This strategy would mathematically be equivalent to a weighted average of rates of transition to disability and mortality due to these other causes. Accounting for calendar trends, enabled us to estimate PYLL with and without disability with greater accuracy compared to methods that assume a constant age-specific prevalence of disease over time.

Our estimates of PYLL were lower than YLLs estimated in the Global Burden of Disease (GBD) study for 2020 and 2021 [11]. This difference in findings is not unexpected, as the GBD methodology calculates YLLs based on the difference between age at death with standard LE at that age. This methodology enables comparability of estimates between diseases and across time and place. However, it may overestimate the true PYLL, as not all individuals expect the ideal LE. Our Markov method allows for a more realistic estimation of expected survival rates based on the health/disability state individuals occupied at time of pre-mature deaths directly or indirectly related to COVID-19. Our results for total PYLL were consistent with those for the European countries in a study which estimated YLL due to the pandemic for 2020 in 81 countries [5]. Our findings for changes in LE explained and not explained by registered COVID deaths are aligned with a previous study which estimated changes in LE in 29 countries for 2020 and 2021 [2], and another study that estimated LE changes in 37 countries for 2020 [3]. Evidence on LE in 2022 and changes in LE and PYLL by disability status have been limited. We found that even though there was a partial recovery in LE since the start of the pandemic by 2022, it did not return to levels expected under pre-pandemic trends in any of the studied countries.

Taken together, our results show a significant portion of PYLL occurred to people capable of leading independent lives without disability, suggesting there may have been a heuristic underestimation of the impact of COVID-19 deaths in previous reports. Increasing PYLL not explained by registered COVID-19 death also pointed to possible long-term indirect impacts of the pandemic, potentially arising from disruptions in healthcare, delivery of services for non-COVID conditions, and unintended consequences of COVID-19 containment measures. The pandemic disproportionately affected men and countries with lower GDP per capita, thus widening the gap in premature mortality between men and women and between higher- and lower-income countries. All these findings highlight a need for better pandemic preparedness in Europe, ideally, as part of a more comprehensive global public health agenda.

## Supporting information

**S1 Appendix. Appendix Methods and Results.** *Appendix Methods.* Table A: Summary of assumptions underlying the Markov models. **Table B:** Numbers and characteristics of study participants in each country. **Table C:** Weighted average of age- and sex-specific baseline prevalence estimates for each health state in the model across all countries. **Fig A:** Weighted

average of transition probabilities to cardiovascular disease, cognitive impairment, dementia, and functional impairment by age, across all countries. **Fig B:** Age- and sex-specific projections for mortality rates by year. **Table D:** COVID-19 mortality rates according to reported numbers of COVID-19 deaths 2020–2022. **Table E:** Relative risks of mortality from each state compared to the general population. **Fig C:** Comparison of mortality rates expected in the absence of the pandemic with estimated actual mortality rates with and without disability over 2020–2022. **Table F:** Proportion of excess deaths in 2020–2022 that was estimated to occur with disability by age group. **Fig D:** Age-standardized all-cause and COVID-19 mortality rates in men and women in 2020 and 2021 in the United Kingdom by disease and disability status, comparing the model outputs with those reported by the Office for National Statistics. **Fig E:** Age-standardized incidence of cardiovascular diseases and Alzheimer's disease across European countries over time. *Appendix Results*. **Fig F**: Person-years of life lost per 1,000 population aged 35 and over, by country, explained and not explained by registered COVID deaths over the years 2020–2022. **Table G:** Person-years of life lost (PYLL) by cause in the population aged 35 and over during the years 2020–2022. **Table H:** Disabled and disability-free person-years of life lost (PYLL) in the population aged 35 and over during the years 2020–2022. **Fig G:** Person-years of life lost with and without disability in the population aged 35 and over, by age-group and sex. **Fig H:** Person-years of life lost per 1,000 population over the years 2020–2022 by vaccination coverage and 2019 Gross Domestic Product per capita. **Fig I:** Change in life expectancy at 35 over time in the decade leading to the pandemic (2009–2019). **Table I:** Life-expectancy at age 35 (LE) with and without disability by country and year. **Fig J:** Life-expectancy at age 35 by gross domestic product per capita, 2019–2022.
(PDF)

## Acknowledgments

This paper uses data from SHARE Waves 1, 2, 4, 5, 6, and 7 (DOIs: https://doi.org/10.6103/SHARE.w1.710, https://doi.org/10.6103/SHARE.w2.710, https://doi.org/10.6103/SHARE.w4.710, https://doi.org/10.6103/SHARE.w5.710, https://doi.org/10.6103/SHARE.w6.710, https://doi.org/10.6103/SHARE.w7.711). The SHARE data collection has been funded by the European Commission, DG RTD through FP5 (QLK6-CT-2001-00360), FP6 (SHARE-I3: RII-CT-2006-062193, COMPARE: CIT5-CT-2005-028857, SHARELIFE: CIT4-CT-2006-028812), FP7 (SHARE-PREP: GA N°211909, SHARE-LEAP: GA N°227822, SHARE M4: GA N°261982, DASISH: GA N°283646) and Horizon 2020 (SHARE-DEV3: GA N°676536, SHARE-COHESION: GA N°870628, SERISS: GA N°654221, SSHOC: GA N°823782) and by DG Employment, Social Affairs & Inclusion through VS 2015/0195, VS 2016/0135, VS 2018/0285, VS 2019/0332, and VS 2020/0313. Additional funding from the German Ministry of Education and Research, the Max Planck Society for the Advancement of Science, the U.S. National Institute on Aging (U01_AG09740-13S2, P01_AG005842, P01_AG08291, P30_AG12815, R21_AG025169, Y1-AG-4553-01, IAG_BSR06-11, OGHA_04-064, HHSN271201300071C, RAG052527A) and from various national funding sources is gratefully acknowledged (see www.share-project.org). The English Longitudinal Study of Ageing, data from which was used in the present study, was developed by a team of researchers based at University College London, NatCen Social Research, the Institute for Fiscal Studies, the University of Manchester and the University of East Anglia. The data were collected by NatCen Social Research. The funding is currently provided by the National Institute on Aging in the US, and a consortium of UK government departments coordinated by the National Institute for Health Research. Funding has also been received by the Economic and Social Research Council.

## Author contributions

**Conceptualization:** Sara Ahmadi-Abhari, Martin J. Shipley, Abbas Dehghan, Mika Kivimaki.

**Data curation:** Sara Ahmadi-Abhari.

**Formal analysis:** Sara Ahmadi-Abhari.

**Funding acquisition:** Sara Ahmadi-Abhari, Abbas Dehghan, Paul Elliott, Mika Kivimaki.

**Methodology:** Sara Ahmadi-Abhari, Piotr Bandosz, Martin J. Shipley, Mika Kivimaki.

**Project administration:** Sara Ahmadi-Abhari.

**Software:** Piotr Bandosz.

**Validation:** Sara Ahmadi-Abhari.

**Visualization:** Sara Ahmadi-Abhari.

**Writing – original draft:** Sara Ahmadi-Abhari, Mika Kivimaki.

**Writing – review & editing:** Sara Ahmadi-Abhari, Piotr Bandosz, Martin J. Shipley, Joni V. Lindbohm, Abbas Dehghan, Paul Elliott, Mika Kivimaki.

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
