## [Editor Report · Decision Letter 0]

3 Sep 2024

Dear Dr Ahmadi-Abhari, 

Thank you for submitting your manuscript entitled "Direct and indirect impacts of the COVID-19 pandemic on life-expectancy and person-years of life lost with and without disability: a systematic analysis for 18 European countries, 2020-2022" for consideration by PLOS Medicine.

Your manuscript has now been evaluated by the PLOS Medicine editorial staff and I am writing to let you know that we would like to send your submission out for external peer review.

Please re-submit your manuscript within two working days, i.e. by Sep 05 2024 11:59PM.

Kind regards,

Syba Sunny, MBBS, MRes, FRCPath

Associate Editor

PLOS Medicine

ssunny@plos.org

---

## [Decision Letter · Decision Letter 1]

4 Oct 2024

Dear Dr Ahmadi-Abhari,

Many thanks for submitting your manuscript "Direct and indirect impacts of the COVID-19 pandemic on life-expectancy and person-years of life lost with and without disability: a systematic analysis for 18 European countries, 2020-2022" (PMEDICINE-D-24-02916R1) to PLOS Medicine. The paper has been reviewed by subject experts and a statistician; their comments are included below and can also be accessed here: [LINK]

As you will see, the reviewers raised several concerns about the data, its analysis and the modelling used. After discussing the paper with the editorial team and an academic editor with relevant expertise, I'm pleased to invite you to revise the paper in response to the reviewers' comments. We plan to send the revised paper to some or all of the original reviewers, and we cannot provide any guarantees at this stage regarding publication.

We ask that you submit your revision by Oct 25 2024 11:59PM. However, if this deadline is not feasible, please contact me by email, and we can discuss a suitable alternative.

Don't hesitate to contact me directly with any questions (ssunny@plos.org). 

Best regards, 

Syba 

Syba Sunny, MBBS, MRes, FRCPath 

Associate Editor

PLOS Medicine

ssunny@plos.org

Comments from the academic editor:

The academic editor thought that the manuscript looked interesting and relevant. She commented that, whilst the reviewers raised many points, none doubted the value of the work. She was supportive of taking your manuscript forward to the next step.

Comments from the reviewers: 

Reviewer #1: The authors attempted to assess the direct and indirect impacts of the COVID-19 pandemic on life expectancy and years of life lost with and without disabilities in 18 European countries, 2020-2022. The authors have been quite successful in achieving their objective. In fact, I only have two comments.

1.- As the authors state on page 3 of the supplementary material, they estimate transition probabilities by fitting Cox proportional hazards regression models.

However, the authors do not test the fulfillment of the proportional hazards assumption. They should test it (both globally and individually for each predictor) and also provide the corresponding Kaplan-Meier curves.

If it is not fulfilled, they should use methods other than the Cox model.

2.- In the Discussion, the authors point out, as a limitation, that 'another source of error is the accuracy of projections for future mortality rates which is inherently unpredictable' and that 'to partly account for uncertainty in projections in mortality and incidence of disease we accompanied our estimates with 95% uncertainty intervals derived from Monte-Carlo simulations'.

I think this is good, but insufficient. The authors should discuss uncertainty in sufficient detail. That is, for example, are the intervals wider in different countries? Are the intervals different over time? etc.

Reviewer #2: Our knowledge of the impact of the pandemic on mortality is still fragmentary, despite a large number of publications on COVID-19. The manuscript aims to provide useful information by providing estimates of person-years of life lost (PYLL) and quantifying them by disability status. The authors use an excess mortality approach with the baseline mortality level estimated using the multi-state Markov chain model. Nevertheless, I have some doubts about the outcome.

1. Relatively old data (last year 2017) is used to estimate disability-free proportion. The assumption that the pandemic didn't change any trends in disability (especially at most affected old ages) is relatively strong and not necessarily correct. 

2. The period used to fit the model (since 1998) might be too long. This would be OK for mid- and long-term forecasts. The short-term forecast used to estimate the expected level of mortality in 2020-2022 is more sensitive to recent changes in mortality trends.

3. In 2020, the number of COVID-19 deaths by country is not comparable between countries. At the beginning of the pandemic, information on deaths from COVID-19 was biased by large differences in approaches to testing SARS-CoV-2 and recoding of COVID-19 as a cause of death. While some countries tend to attribute to COVID-19 all or nearly all deaths of those with positive tests for the virus, others applied more conservative approaches with an emphasis on pre-existing co-morbidities. 

4. The importance of GDP seems to be overestimated by the authors. For example, several publications noted an East-West gradient in pandemic losses in 2021. It corresponds to the same gradient in mortality. The correlation between GDP and mortality was described by Presto many years ago but it doesn't explain why Western countries had lower losses in 2021. In addition, the correlation between GDP and losses doesn't work in 2020. In this light, factors such as vaccination coverage and trust in science and government seem to be more important.

Finally, the description of the model in the supplementary material is not detailed enough and is poorly structured. It is impossible to evaluate the model and its possible limitations. 

Reviewer #3: 1- While the use of a Markov model is appropriate for modeling longitudinal transitions, could the authors provide more justification for choosing this model over other potential approaches (e.g., Cox proportional hazards models for time-to-event data or microsimulation models)? Even though the Markov model captures state transitions well, a brief explanation for this choice, particularly given the model's memoryless property (i.e., transitions depend only on the current state and not on past history), would be beneficial.

2- The manuscript outlines the assumptions of the model clearly. However, I suggest including a brief discussion of the limitations these assumptions might impose, especially regarding the transition probabilities derived from datasets that predate the pandemic.

3- The use of population numbers and mortality rates from sources like STMF and the WHO is appropriate, and the explanation of these inputs is clear. However, it would be helpful to explain more clearly how country-specific variations in data availability and quality were handled, as these differences could affect the robustness of the model's outputs.

4- The use of ELSA and SHARE data to estimate transition probabilities is well justified. However, the manuscript would benefit from including sensitivity analyses that account for variability in these inputs, as the reliability of data may differ between countries.

5- The authors mention Monte Carlo simulations to generate uncertainty intervals, but providing a more detailed explanation of how variation in transition probabilities across different countries was managed would enhance the transparency of the model.

6- The use of Monte Carlo simulations to estimate uncertainty intervals is appropriate given the model's complexity and reliance on probabilistic transitions. However, it could be helpful to explain why 500 iterations were chosen and whether higher or lower numbers of iterations were tested to assess their impact on the stability of the uncertainty estimates.

7- The estimation of PYLL (person-years of life lost) with and without disability is a strength of the analysis. However, given the lack of direct data on disability-related mortality in many countries, the assumptions used to estimate these inputs should be better justified. It would be useful to explain why hazard ratios from the UK were applied across other countries.

8- The two extreme scenarios—either assuming all reductions in non-COVID deaths were real or assuming they resulted from misclassification as COVID-19 deaths—offer a reasonable approach to managing uncertainty in cause of death. However, the analysis could benefit from more discussion of the likely midpoint between these extremes, as well as an exploration of intermediate assumptions and how they might impact the final results.

9- The incorporation of time-specific transition probabilities based on pre-pandemic trends enhances the dynamic nature of the model. However, since the COVID-19 pandemic significantly disrupted healthcare access, it would be useful to discuss how these disruptions were accounted for in the model beyond mortality rates (e.g., access to preventive care or management of chronic diseases).

10- Conducting sensitivity analyses using Monte Carlo simulations adds robustness to the findings, but additional information on the distribution assumptions for the parameters would be helpful. For example, are transition probabilities assumed to follow a normal or log-normal distribution? 

11- The authors mention that the methods and assumptions used in this Markov model have been validated for the UK. It would strengthen the manuscript to elaborate on any cross-country validation performed to ensure the assumptions are valid across the 18 European countries, given the variations in healthcare systems, demographic structures, and responses to COVID-19. If such validation was not feasible, this limitation should be explicitly stated.

12- The manuscript could benefit from a more explicit discussion of how missing data (if any) were handled in the model. Given that longitudinal data sources like ELSA and SHARE may contain incomplete information, it is important to explain how this was addressed (e.g., through imputation, exclusion, or other methods).

13- Given that the study spans the period from 2020 to 2022, which included multiple waves of COVID-19 and the emergence of different variants (e.g., Alpha, Delta, Omicron), it would be helpful to discuss how, if at all, these evolving factors were considered in the modeling. Was the pandemic treated as a uniform event, or did time-varying effects of different variants impact mortality and healthcare access?

14- The model assumes discrete health states with transitions between them. It would be useful to clarify if and how interactions between health states (e.g., cardiovascular disease co-occurring with dementia) were handled in the model. Can individuals occupy multiple health states simultaneously, and how might this affect transitions and mortality risk?

15- The manuscript mentions the indirect impacts of the pandemic due to disruptions in healthcare services. It would be beneficial to expand on whether other external shocks (e.g., changes in policy or healthcare infrastructure strain) were modelled, particularly in relation to non-COVID causes of death. For example, were the lagged effects of disruptions (e.g., delayed cancer screenings) on long-term mortality, which may not fully manifest within the 2020-2022 window, considered?

16- The model stratifies by age groups (35+, 65+, etc.), but it might be helpful to discuss whether age-cohort effects (i.e., differences between generations) were considered. For instance, older cohorts may have different health risks and exposures compared to younger cohorts. Even though the model is time-based, it could be valuable to discuss how aging dynamics and cohort-specific vulnerabilities were factored in or could affect the results.

17- The mention of a bespoke R package developed by one of the authors is a positive step toward reproducibility. However, it would be useful to clarify whether this code will be made publicly available, as doing so would enhance the transparency of the modeling process.

18- The supplementary material contains valuable information such as tables and figures that aid in understanding the modeling assumptions and results. It is important to ensure that the main manuscript clearly directs readers to these tables and figures, especially for those who may not read the supplementary material in full detail.

---

* Please upload any figures associated with your paper as individual TIF or EPS files with 300dpi resolution at resubmission; please read our figure guidelines for more information on our requirements: http://journals.plos.org/plosmedicine/s/figures. While revising your submission, please upload your figure files to the PACE digital diagnostic tool, https://pacev2.apexcovantage.com/. PACE helps ensure that figures meet PLOS requirements. To use PACE, you must first register as a user. Then, login and navigate to the UPLOAD tab, where you will find detailed instructions on how to use the tool. If you encounter any issues or have any questions when using PACE, please email us at PLOSMedicine@plos.org.

* Please ensure that the paper adheres to the PLOS Data Availability Policy (see http://journals.plos.org/plosmedicine/s/data-availability). For data residing with a third party, authors are required to provide instructions with contact information (web or email address) for obtaining the data. Please also note that a study author cannot be the contact person for any data. Please direct readers to a non-author institutional point of contact, such as a data access or ethics committee. Providing a durable point of contact ensures data will be accessible even if an author changes email addresses, institutions, or becomes unavailable to answer requests.

* Thank you got providing an Author Summary. Please note that this is expected to be short, non-technical and the aim is to make your research accessible to a wide audience that includes both scientists and non-scientists. The Author Summary should immediately follow the Abstract in your revised manuscript. Ideally each sub-heading should contain 2-3 single sentence, concise bullet points containing the most salient points from your study. We ask that you revise the section entitled ‘What Did the Researchers Do and Find?’ to be shorter and use less technical language. Also, in the final bullet point of 'What Do These Findings Mean?', please include the main limitations of the study in non-technical language. Please see our author guidelines for more information: https://journals.plos.org/plosmedicine/s/revising-your-manuscript#loc-author-summary.

FIGURES AND TABLES

SUPPLEMENTARY MATERIAL

REFERENCES

OBSERVATIONAL STUDIES

* Please ensure that the study is reported according to the appropriate reporting guideline (available from: https://www.equator-network.org/reporting-guidelines) and include the completed checklist as Supporting Information. Please add the following statement, or similar, to the Methods: "This study is reported as per the [insert reporting guideline name and abbreviation here] (S1 Checklist)." When completing the checklist, please use section and paragraph numbers, rather than page numbers. 

* For all observational studies, in the manuscript text, please indicate: (1) the specific hypotheses you intended to test, (2) the analytical methods by which you planned to test them, (3) the analyses you actually performed, and (4) when reported analyses differ from those that were planned, transparent explanations for differences that affect the reliability of the study's results. If a reported analysis was performed based on an interesting but unanticipated pattern in the data, please be clear that the analysis was data driven. 

* Please state in the Methods section whether the study had a prospective protocol or analysis plan. If a prospective analysis plan (from your funding proposal, IRB or other ethics committee submission, study protocol, or other planning document written before analyzing the data) was used in designing the study, please include the relevant document(s) with your revised manuscript as a Supporting Information file to be published alongside your study and cite it in the Methods section. A legend for this file should be included at the end of your manuscript. If no such document exists, please make sure that the Methods section transparently describes when analyses were planned, and when/why any data-driven changes to analyses took place. Changes in the analysis, including those made in response to peer review comments, should be identified as such in the Methods section of the paper, with rationale.

---

## [Decision Letter · Decision Letter 2]

10 Jan 2025

Dear Dr. Ahmadi-Abhari,

Thank you very much for re-submitting your manuscript "Direct and indirect impacts of the COVID-19 pandemic on life-expectancy and person-years of life lost with and without disability: a systematic analysis for 18 European countries, 2020-2022" (PMEDICINE-D-24-02916R2) for review by PLOS Medicine.

I have discussed the paper with my colleagues and the academic editor and it was also seen again by 2 of the original reviewers (including the statistician). I am pleased to say that provided the remaining editorial and production issues are dealt with we are planning to accept the paper for publication in the journal.

[LINK]

We look forward to receiving the revised manuscript by Jan 17 2025 11:59PM. But please do let us know if you need more time.

Sincerely,

Syba

Syba Sunny, MBBS, MRes, FRCPath

Senior Editor 

PLOS Medicine

plosmedicine.org

Requests from Editors:

Thank you for submitting your revised manuscript. As you can see, the reviewers were positive about your amendments and we are pleased to be moving forward with your submission. One of the reviewers (Reviewer 3) provided some additional suggestions. In your next revision, we ask that you kindly provide a point-by-point response to these comments as well as the editorial requests outlined below. 

(1) Data Availability Statement

Thank you for providing a Data Availability Statement. However, we kindly ask that you provide some extra information for the benefit of our readers. For data residing with any third party, authors are required to provide instructions with contact information (web or email address) for obtaining the data. Please provide this information with respect to the ELSA and SHARE studies. Please also note that a study author cannot be the contact person for any data. Please direct readers to a non-author institutional point of contact, such as a data access or ethics committee. Providing a durable point of contact ensures data will be accessible even if an author changes email addresses, institutions, or becomes unavailable to answer requests.

(2) Author Summary:

We ask that you kindly amend your Author Summary a little more. Please use bullet points. (Ideally each sub-heading should contain 2-3 single sentence, concise bullet points containing the most salient points from your study.) Please provide an expansion of GDP at first use in the Author Summary. The section entitled ‘What Did the Researchers Do and Find?’ would benefit from being shortened further. We also ask if the authors could revise the first 2 sentences of the last section to include a more concrete example of a study limitation. It may be best to remove the first 2 sentences entirely, start this section with the sentence ‘ The increasing PYLL…’ and write about a study limitation towards the end of this section instead. The authors could say something along the lines of ‘Limitations of this study include the lack of mortality data by disability status that was available during the pandemic and the subsequent reliance on estimates.’

Comments from Reviewers:

Reviewer #1: The authors have responded very well not only to my comments but also to those of the other reviewers. In addition, they have included many of them in the new version of the manuscript. I have no further comments.

Reviewer #3: Thank you for thoroughly addressing the comments and suggestions I provided in my initial review. The revisions have significantly improved the clarity, depth, and methodological rigor of the manuscript. I appreciate the detailed responses to each point, particularly the explanations regarding the modelling choices, data handling, and statistical analyses. Below are some additional comments that, if addressed, could further strengthen the manuscript.

- While the Markov model uses age- and sex-specific strata, the authors could discuss how granularity in the inputs (e.g., socioeconomic or regional differences within countries) might impact the outputs.

- While the authors chose 500 iterations for Monte Carlo simulations, it may be useful to present results of convergence diagnostics or explore whether higher iterations (e.g., 1000) would meaningfully change uncertainty intervals.

-Provide additional justification for the use of normal distributions for transition probabilities. For some events (e.g., rare transitions), log-normal or beta distributions might be more suitable. Explain why normal distributions were deemed appropriate in this context.

- Given that pre-pandemic data trends were assumed to be linear or log-linear, a brief explanation of how non-linear patterns (if present) were tested and ruled out would strengthen the methodology.

[LINK]

---

## [Decision Letter · Decision Letter 3]

27 Jan 2025

Dear Dr Ahmadi-Abhari, 

On behalf of my colleagues and the Academic Editor, Professor Mirjam Kretzschmar, I am pleased to inform you that we have agreed to publish your manuscript "Direct and indirect impacts of the COVID-19 pandemic on life-expectancy and person-years of life lost with and without disability: a systematic analysis for 18 European countries, 2020-2022" (PMEDICINE-D-24-02916R3) in PLOS Medicine.

PRESS

Sincerely, 

Syba

Syba Sunny, MBBS, MRes, FRCPath 

Associate Editor 

PLOS Medicine